# Integration of Single-Cell Analysis and Bulk RNA Sequencing Data Using Multi-Level Attention Graph Neural Network for Precise Prognostic Stratification in Thyroid Cancer

**DOI:** 10.3390/cancers17142411

**Published:** 2025-07-21

**Authors:** Langping Tan, Zhenjun Huang, Yongjian Chen, Zehua Wang, Zijia Lai, Xinzhi Peng, Cheng Zhang, Ruichong Lin, Wenhao Ouyang, Yunfang Yu, Miaoyun Long

**Affiliations:** 1Guangdong Provincial Key Laboratory of Malignant Tumor Epigenetics and Gene Regulation, Department of Thyroid Surgery, Department of Medical Oncology, Sun Yat-sen Memorial Hospital, Sun Yat-sen University, Guangzhou 510120, China; tanlp7@mail.sysu.edu.cn (L.T.); huangzhj79@mail2.sysu.edu.cn (Z.H.); laizj6@mail2.sysu.edu.cn (Z.L.); pxinzh@mail.sysu.edu.cn (X.P.); zhangch339@sysu.edu.cn (C.Z.); auyeung3@mail2.sysu.edu.cn (W.O.); 2Dermatology and Venereology Division, Department of Medicine Solna, Center for Molecular Medicine, Karolinska Institutet, 171 77 Stockholm, Sweden; chenyj326@mail3.sysu.edu.cn; 3UMedEVO and UMedREVO Artificial Intelligence Technology (Guangzhou) Co., Ltd., Guangzhou 510275, China; wangzehua@uic.edu.cn; 4Faculty of Innovation Engineering, Macau University of Science and Technology, Taipa, Macao, China; r130233046@mail.uic.edu.cn; 5Guangdong Provincial Key Laboratory of Cancer Pathogenesis and Precision Diagnosis and Treatment, AI Big Data Laboratory, Shenshan Medical Center, Memorial Hospital of Sun Yat-sen University, Shanwei 516600, China; 6Institute for AI in Medicine and Faculty of Medicine, Macau University of Science and Technology, Taipa, Macao, China; 7Department of Breast Surgery, The First Affiliated Hospital, Jinan University, Guangzhou 510632, China; 8Department of Thyroid Surgery, The Sixth Affiliated Hospital of Sun Yat-sen University, Sun Yat-sen University, Guangzhou 510655, China

**Keywords:** thyroid cancer, Tcell, single-cell transcriptomic analysis, tumor immune microenvironment, disease-free survival

## Abstract

This study investigates the role of T cells in thyroid cancer, a major factor in determining the prognosis of the disease. Researchers aim to improve the accuracy of predicting patient outcomes by analyzing RNA-seq data from both bulk and single-cell samples. The study focuses on how different T-cell subtypes contribute to thyroid cancer and uses advanced machine learning techniques to create a model that predicts disease-free survival. The findings highlight the importance of T-cell-based genetic features in predicting the prognosis of thyroid cancer patients and could lead to better treatment strategies and enhanced understanding of the tumor’s response to immunotherapy.

## 1. Introduction

Thyroid cancer is the most prevalent malignancy among 16–33 years old, and it is ninth in global cancer incidence [1]. According to cancer statistics, it has been estimated that the number of new thyroid cancer cases is 3% in females [2,3,4]. Research also reveals lymph node metastasis as a significant factor for poor prognosis in thyroid cancer [5]. The rate of overdiagnosis in thyroid cancer varies between 60% and 90%, depending on cancer subtype and population. Papillary thyroid carcinoma (PTC), the most common subtype, is particularly prone to overdiagnosis due to its indolent nature. Overdiagnosis is more common in populations undergoing extensive screening, such as those with widespread ultrasound use for early detection of asymptomatic thyroid nodules [6,7,8]. Additionally, overdiagnosis rates are higher in younger individuals and women, as well as in regions with advanced diagnostic technologies, raising concerns about unnecessary treatment for clinically insignificant cancers [6,7,8]. This variability highlights the need for careful consideration of overdiagnosis risks in thyroid cancer screening and treatment protocols.

However, solely depending on it for prognosis prediction is challenging in clinical practice. Thus, there is an urgent need for novel biomarkers that can predict thyroid cancer prognosis more conveniently and reliably.

Single-cell transcriptome sequencing has been a rapidly evolving technique over the past decade. It can properly detect and study cell clusters and associated gene functions in complex mixtures. A previous study reported the tumor origin and malignant evolution of thyroid cancer through single-cell sequencing technology and revealed in depth the tumor heterogeneity of thyroid cancer [9]. However, the mechanism of emergence and development of thyroid cancer has not been clarified. Thus, scRNA-seq technology offers unprecedented opportunity to assess the mechanisms of thyroid Cancer. The immune microenvironment, specifically T cells, plays a significant role in thyroid cancer. While T cells can attack cancer cells, thyroid cancer can also modify these cells to promote its growth. Hence, understanding the behavior of T cells within thyroid tumors is crucial. The immune microenvironment offers a novel and practical way to predict thyroid cancer prognosis [10]. All in all, scRNA-seq allows us to investigate the immune microenvironment in greater detail. By studying these interactions, we can identify patterns that predict thyroid cancer progression.

Deep learning has achieved impressive performance on many bioinformatics tasks compared to traditional machine learning algorithms [11]. Wang et al. conducted a multi-omics clustering analysis on 348 thyroid cancer samples, identifying three molecular subtypes with different prognostic outcomes and therapeutic implications [12]. In recent years, deep learning algorithms have been demonstrated to predict immunotherapy efficacy by analyzing tumor immune infiltration and characterizing the tumor ecosystem. The strength of artificial intelligence lies in its ability to integrate complex data for patient diagnosis. To address this, we utilize the multi-level attention graph neural network (MLA-GNN), a graph neural network architecture that has demonstrated superior performance in tasks involving omics data integration. MLA-GNN excels at capturing both global and local relationships within the data by applying multi-level attention mechanisms, making it particularly well-suited for integrating heterogeneous data sources, such as single-cell and bulk genomic data. By leveraging the complex statistical structure of multiple hidden layers and nonlinear activation functions, deep learning algorithms like MLA-GNN enable us to develop a method for integrating genomic features to identify thyroid cancer patients from risk-matched controls and predict their disease-free survival (DFS) with high precision. Therefore, we exploit the complex statistical structure of multiple hidden layers and nonlinear activation functions that deep learning algorithms such as MLA-GNN have in order to develop a method for integrating genomic features to identify thyroid cancer patients from risk-matched controls and to predict the DFS of thyroid cancer precisely.

This study used scRNA-seq and transcriptome sequencing to analyze T-cell composition in the thyroid cancer microenvironment. It emphasized Treg cells’ role in cancer progression and the DFS profile of patients through multi-omics analysis. Our research reveals that high Treg cell expression and elevated MHC-II expression are found in tumors. Treg cells, showing enhanced antigen presentation functions, are also involved in tumor metabolism. Notably, Treg cells can inhibit anti-tumor immunity by suppressing cytotoxic T lymphocytes, and their elevated MHC-II expression may further modulate antigen presentation to promote immune tolerance. In addition, Treg cells exhibit metabolic reprogramming that supports their proliferation and suppressive function, which collectively contributes to tumor immune evasion and progression. Furthermore, we developed a prognostic model based on genomic features with high accuracy for predicting thyroid cancer prognosis.

### Study Hypothesis and Objective

In this study, our primary hypothesis was that integrating RNA-seq and scRNA-seq data could identify key T-cell-related genes that are critical for thyroid cancer prognosis. We aimed to develop a robust prognostic model that can predict DFS and help stratify patients into high-risk and low-risk groups. By analyzing these data, we sought to enhance the understanding of the tumor microenvironment, particularly focusing on T-cell-related mechanisms that influence disease progression. Ultimately, the objective was to provide a model that can guide future treatment strategies and identify potential therapeutic targets for thyroid cancer.

## 2. Materials and Methods

### 2.1. Acquisition of Raw Data

The scRNA-seq data for thyroid cancer included 15 thyroid cancer samples (seven primary tumor tissues, six peritumor tissues, and two subcutaneous metastases), and 158,557 eligible cells were sourced from the GEO database (GSE184362). From The Cancer Genome Atlas (TCGA) database, we sourced a wealth of information, including transcriptomic data and single nucleotide variants (SNVs). Alongside this genetic information, we gathered corresponding clinical information pertaining to thyroid carcinoma (THCA), providing us with a multifaceted perspective of the disease at a molecular level. For the transcriptomic data, we excluded samples lacking survival data and outcome status, resulting in the final inclusion of 489 tumor samples. And then the patients were categorized into two groups with an 8:2 ratio. These two cohorts served as the training and validation sets, respectively, each comprising THCA RNA expression patterns. All expression data were converted into transcripts per million (TPM) format followed by log2 transformation, which adjusts for sequencing depth and gene length, allowing for the accurate comparison of gene expression levels across samples in bulk RNA-seq datasets. Batch effects across TCGA samples were corrected using the ‘ComBat’ function from the sva package before downstream analyses, including immune cell deconvolution by CIBERSORT. Following this, batch effects were eliminated using the ‘ComBat’ function from the sva R package. To assess the effectiveness of correction, we generated PCA and UMAP plots before and after batch adjustment (Appendix A), which showed improved integration and reduced sample-specific clustering. Before proceeding with the analysis, all data underwent a conversion process with Log2. With these preparations complete, a total of 608 genes, each with associated relevance scores, were selected for subsequent investigation.

### 2.2. Processing of scRNA-Seq Data and Cell Annotation

The scRNA-seq data were evaluated for accuracy using the “Seurat” R package. We implemented a filter to retain only the genes expressed in a minimum of three single cells, and cells that expressed between 200 and 4000 genes. Moreover, cells having more than 10% of mitochondrial genes were excluded to ensure the quality of the scRNA-seq data. A total of 158,557 eligible cells were selected for further analysis. The data of these remaining cells were normalized using the “LogNormalization” method, which is commonly applied in scRNA-seq analysis to address the sparsity and variability of gene expression across cells. This method normalizes each cell by its total expression, multiplies by a scale factor, and applies log-transformation to stabilize variance. The “FindVariableFeatures” function helped in identifying the top 1000 highly variable genes post-normalization. Given the fact that the data were extracted from multiple samples, the batch effects that could potentially disrupt the subsequent analysis were eliminated using the “FindIntegrationAnchors” function of the canonical correlation analysis (CCA) approach. The “IntegrateData” and “ScaleData” functions were then used for accurate integration and scaling of the data. Principal component analysis (PCA) was deployed for dimensionality reduction to determine anchor points. We utilized the uniform manifold approximation and projection (UMAP) algorithm to examine the top 20 principal components (PCs) for the identification of meaningful clusters. We succeeded in obtaining 18 cell clusters using the “FindNeighbors” and “FindClusters” functions, with a resolution of 0.4, which was selected after testing multiple values (0.2–1.0) and evaluating cluster separation and marker gene expression. UMAP visualization was performed using the top 20 principal components (PCs), selected based on ElbowPlot inspection and the proportion of variance explained, ensuring a balance between data complexity and noise reduction. We assessed the heterogeneity along the cell cycle based on the cell cycle markers provided in the “Seurat” package. The “FindAllMarkers” function of the “Seurat” package was utilized to discern DEGs in each cluster. The genes acting as markers for each cluster were identified using adjusted *p*-values (FDR) < 0.05 and log2(fold change) > 0.25 as selection criteria. Multiple testing correction was performed using the Benjamini–Hochberg method implemented in the Seurat and limma packages. Cell types were then carefully confirmed based on the canonical marker genes of each cluster [13].

### 2.3. Utilization of AUCell

We utilized the “AUCell” R package to analyze the activity levels of genesets in scRNA-data and assign related-scores to each cell lineage. The distribution of these scores was visualized using “ggplot2 R” software (3.5.1), allowing us to calculate the scores of metabolic pathways between tumor group and normal group, based on the median values of the AUC score. To delve into the heterogeneity of various biological processes, we utilized the “GSVA” package to conduct a gene set variation analysis (GSVA). This analysis was performed for both the high- and low-Sphingolipid-AUC groups with the aim of identifying the biological pathways enriched in each group. The noteworthy pathways discovered during this analysis were presented visually in the form of a bar chart.

### 2.4. Enhanced Analysis of Cell–Cell Communication

To explore cell–cell communication in detail, we used the CellChat (1.1.0) platform, a comprehensive and publicly accessible database that provides data on ligands, receptors, cofactors, and their interactions. CellChat is a versatile, user-friendly toolkit that facilitates the comprehensive exploration of intercellular communication and allows the creation of detailed communication atlases. We used CellChat to calculate expression levels for a set of coding genes across all transcriptomes, relative to the total read mapping. This approach allowed us to account for all potential factors contributing to cell–cell communication. To refine our data, we averaged the expression values within each single-cell cluster or sample. This approach allowed us to identify both broad patterns and subtle variations in cell–cell communication across clusters or samples, providing a deeper understanding of the complex interactions.

### 2.5. Gene Co-Expression Computation

The initial step in utilizing MLA-GNN for omics analysis involves encoding the omic data of each patient into a graph representation, defined by an input feature matrix *V* and an edge matrix *E*. Assuming that each patient is represented by *K* genes, the feature matrix $V^{K\times 1}$ corresponds to a graph with *K* nodes, where each node represents the expression level of one gene. Gene co-expression analysis was conducted through weighted gene co-expression network analysis (WGCNA) to compute the edge matrix $E^{K\times K}$. By employing WGCNA, the network’s edges are determined based on the strength of co-expression relationships between genes, capturing their co-regulation patterns. Genes work interactively to regulate biological processes [14,15,16,17]; thus, the correlated co-functional gene modules (or activated pathways) can better reflect the disease status and biological processes. Therefore, the co-functional gene modules connected on the co-expression graph could be processed to extract high-level features of the biological gene modules, thus boosting the performance of disease predictions.

### 2.6. MLA-GNN Model

The MLA-GNN (multi-layer attention graph neural network) architecture is designed to effectively model graph-structured data by combining multiple layers of graph attention mechanisms with a sequence of fully connected layers for feature refinement and classification. The network begins with three stacked graph attention layers, each with multiple attention heads (4, 3, and 4, respectively), allowing it to capture diverse and hierarchical node relationships. These layers apply attention mechanisms to weigh the importance of neighboring nodes, progressively refining node embeddings from lower-level to higher-level representations. Following the attention layers, three fully connected layers project these embeddings into lower-dimensional abstract feature spaces (64, 48, and 32 dimensions), enhancing the model’s ability to learn discriminative features relevant to the downstream classification task. The final classification layer maps these representations to the output space defined by the number of target classes.

Training proceeds over a user-defined number of epochs (default 200) with a batch size of 8, using optimizers such as Adam or Adagrad, as specified by the optimizer_type argument. The default learning rate is 1 × 10^−4^, with weight decay set to 5 × 10^−4^ for L2 regularization, helping to reduce overfitting. Learning rate scheduling is controlled by lr_policy, with options like ‘linear’ to progressively decrease the learning rate across epochs. The training loss typically combines a classification loss (e.g., cross-entropy), a regularization term scaled by lambda_reg, and optionally survival or negative log-likelihood losses (lambda_cox, lambda_nll) depending on the task. Dropout is applied at a rate of 0.2 to prevent overfitting, and LeakyReLU with α = 0.2 is used within attention mechanisms to introduce nonlinearity. Training also supports early stopping via a patience mechanism and random seeding (--seed 10) for reproducibility. During each iteration, the model processes node features and graph structure, computes attention-based embeddings, applies feature transformations through fully connected layers, generates predictions, computes loss, and updates parameters through backpropagation.

### 2.7. Prognosis Prediction Using MLA-GNN Model

In this study, we propose utilizing the MLA-GNN model to establish a prognosis prediction system in the medical field, specifically for oncology [18]. To achieve this, we first represent the data of each patient as a graph, known as the gene co-expression graph. This graph is constructed using the WGCNA method, allowing us to extract advanced features from biological gene modules, and ultimately yielding the gene co-expression graph [19]. Through this approach, we enhance the predictive performance for disease prognosis [20,21].

Next, we input the gene co-expression graph into the MLA-GNN model, effectively combining its features through continuous fully connected layers. This process facilitates the encoding of essential information inherent in the graph structure [22]. Subsequently, we conduct disease classification and survival prediction tasks in the prediction module.

To evaluate the performance of our novel model, we employ the receiver operating characteristic (ROC) function to estimate its accuracy in prognostic predictions. The ROC curve graphically illustrates the trade-off between the true positive rate and false positive rate, offering a comprehensive assessment of the model’s predictive ability. A higher area under the curve (AUC) of the ROC curve indicates stronger predictive capabilities.

### 2.8. Profiling of Immune Cell Subpopulations

CIBERSORT [23] employs a quantitative approach to predict the proportions of different immune cell types present in complex cellular mixtures. This process involves a rigorous analysis of expression profiles, enabling the identification of immune cell subpopulations with high precision. We employ the Wilcoxon test to discern significant differences in immune cell proportions between low- and high-risk populations. The integrated use of CIBERSORT and statistical analyses offers valuable insights into the role of the immune microenvironment in disease progression.

### 2.9. Genetic Alterations and Tumorigenesis Through Mutation Analysis

Mutation analysis is a critical step in comprehending the genetic alterations occurring in the organism and their potential connection to tumorigenesis. Mutation detection was conducted using the R software package “maftools” (2.25.10), which offers robust functionalities for the analysis of mutation annotation format (MAF) files.

### 2.10. Enrichment Analysis to Unveil Biological Significance in Genomic Data

A differential analysis was performed on 489 patients using the “limma” package with *p* value < 0.05, and |Log2FC| > 1 were designated as DEGs. The heatmaps and volcano plots of the DEGs were visualized using the “ggplot2” and “pheatmap” packages, respectively. Enrichment analysis was performed using the clusterProfiler package, and Benjamini–Hochberg FDR correction was applied to account for multiple comparisons. Only terms with adjusted *p*-value < 0.05 were considered significantly enriched. In this study, we performed gene ontology (GO) and Kyoto Encyclopedia of Genes and Genomes (KEGG) pathway enrichment analysis using the R software package “clusterProfiler” (4.16.0) [24]. This powerful tool allows us to gain deeper insights into the functional significance of our gene set in the context of cancer research.

### 2.11. Subtype Clinical Feature Analysis

Samples exhibiting various clinical-pathological features were classified into subtypes based on age (>55 or ≤55 years), sex, stages, T stage, N stage, and M stage. Within each subtype, cancer samples were divided into two risk groups (high-risk and low-risk). The distribution of clinical-pathological features between subtypes was evaluated using the Kruskal–Wallis test or Wilcoxon rank test. To better understand the correlation between clinical-pathological features and survival rates, stratified survival analysis was performed for high-risk and low-risk populations. This comprehensive approach aids in the nuanced evaluation of disease progression and prognosis, considering the diversity of patient characteristics and disease stages.

## 3. Results

We selected a total of 489 and 11 patients with thyroid cancer from the TCGA and GEO databases, respectively. Detailed clinical characteristics of these patients can be found in Table 1. The study’s flowchart is displayed in Figure 1.

### 3.1. Characterization of the Tumor Microenvironment in Thyroid Tissues Through Identifying Main Clusters

In order to systematically examine TME, our initial approach was to analyze the scRNA-seq data of fifteen thyroid tissues—comprising seven primary tumor tissues, six peritumor tissues, and two subcutaneous metastases—sourced from the GSE184362 dataset [25], using the 10X Genomics platform (Figure 2A). Referencing prior research [10,26], we classified the seven primary tumor samples and two subcutaneous metastases as ‘tumor tissue’. The remaining six peritumor tissues were classified as ‘normal tissue’ based on their spatial separation from the tumor mass and histological annotation in the original dataset. We then employed a machine-learning algorithm specifically for single-cell annotation prediction, conducting stringent quality control filters and dimensionality reduction. Following these preliminary data preprocessing steps, we were able to preserve 122,979 high-quality single-cell measurements in the dataset, with no significant batch effects observed within the thyroid tissue samples (Figure 2A). Using differential expression analysis and known marker genes for our anticipated major cell types, we manually annotated the graph-based clusters (Figure 2B and Appendix A). Cell types were identified using known marker genes such as CD3D, CD3E, CD3G, and CD247 for T cells; CD79A, CD79B, IGHD, and IGHM for B cells; LYZ, S100A8, S100A9, and CD14 for myeloid cells; TG, EPCAM, KRT19, and KRT18 for thyroid cells; COL1A1, COL1A2, COL3A1, and ACTA2 for fibroblasts; and CDH5, PECAM1, VWF, and CD34 for endothelial cells. This method led us to the identification of six distinct cell clusters, comprised of 18,720 B cells, 36,443 T cells, 11,856 myeloid cells (monocytes, macrophages, and dendritic cells, excluding granulocytes), 48,322 thyroid cells, and a smaller representation of endothelial cells and fibroblasts (Figure 2B,C and Appendix A).

Applying average marker expression, we performed graph-based clustering to discern the specific cell types present (Figure 2D). Upon comparing cell proportions between the tumor and normal tissues, we observed a relative paucity of T cells and B cells in the tumor samples, while myeloid cells, thyroid cells, and fibroblasts were enriched. This observation may reflect an immune exclusion phenotype or stromal barrier characteristic of PTC, in which lymphocyte infiltration into tumor nests is limited.

### 3.2. T-Cell Heterogeneity and Metabolic Landscape in Thyroid Tumor Microenvironment Revealed by Single-Cell RNA Sequencing

To enhance our comprehension of the functional heterogeneity of T lymphocytes infiltrating both normal and neoplastic tissues, a graph-based clustering approach was utilized to divide T-cell clusters into four main categories: Natural Killer (NK) cells, CD8^+^ T cells, CD4^+^ T cells, and regulatory T cells (Tregs) (Figure 3A and Appendix A). The expression of marker genes for individually discerned T-cell populations and phenotypic states was represented utilizing Uniform Manifold Approximation and Projection (UMAP) feature plots (Figure 3B and Appendix A). Lastly, we scrutinized the proportions of T-cell categories in both normal and neoplastic tissues. A relative decrease in the proportion of CD4^+^ cells in tumor tissues was observed, whereas the proportions of CD8^+^ cells, NK cells, and Treg cells experienced various increments (Figure 3C).

Next, we performed a differential analysis to determine the DEGs of T cells between tumor and normal tissues, and a total of 658 DEGs were selected for follow-up studies. Volcano plots illustrated the global differential expression between the cohorts. In total, 469 genes exhibited higher expression levels in tumor tissues compared to normal tissues, including S100A6 and S100A4 (Appendix A). The upregulation of certain genes, such as MT1G and RPL41, typically correlated with T-cell activation, was also observed in neoplastic tissues. Gene set enrichment analysis corroborated the elevated expression of T-cell activation pathways, thus paralleling the results of the differential gene analysis (Figure 3D). In addition, the DEGs of NK cells, CD8^+^ T cells, CD4^+^ T cells, and Treg cells between tumor and normal tissues were also performed (Appendix A).

Subsequently, we depicted the metabolic landscape of both tumor and normal tissues, specifically within the glycolytic pathway. We enlisted the Kyoto Encyclopedia of Genes and Genomes (KEGG) database to analyze CD4^+^ T cells, CD8^+^ T cells, Treg cells, and NK cells in both normal and cancerous tissues. Our study results further underscored that Treg cells and NK cells in tumor tissues exhibit enhanced metabolic activity in these pathways (Figure 3E and Appendix A).

According to the above analysis, metabolism has been established to influence the phenotype and function of immune cells within TME, which includes T cells [27]. In order to gain an unbiased, thorough understanding of the metabolic activity within T-cell subpopulations in the TME, we conducted Ucell analysis, focusing on the pentose phosphate, glycolysis, and oxidative phosphorylation metabolic processes for CD4^+^ T cells, CD8^+^ T cells, Treg cells, and NK cells in the tumor and in normal tissues (Figure 3F). Our findings indicated a major enrichment of the glycolytic pathway in Treg cells within tumor tissues (Appendix A).

### 3.3. Comparative Intercellular Communication Analysis in Thyroid Cancer

To statistically assess differences in cell–cell communication patterns in thyroid cancer, we performed an integrated comparative analysis using the *CellChat* framework based on the merged single-cell dataset. This approach allowed us to model the global communication networks across both tissue types simultaneously, and to quantitatively compare the interaction number, strength, and signaling pathway activity between groups. The number of significant ligand–receptor interactions varied markedly (Figure 4A). It displayed an overall increase in intercellular interactions, particularly among immunosuppressive populations such as CD4^+^ T cells, CD8^+^ T cells, NK cells, and Treg cells, reflecting a more complex and potentially immunoevasive communication network. It also demonstrates increased total interaction strength among specific cell subsets between NK cells and other T cells in tumor tissues, suggesting that not only the number but also the intensity of cellular communication is altered in the tumor microenvironment (Figure 4B).

The relative information flow of canonical signaling pathways has also been evaluated (Figure 4C). It has also been found that NK cells interact the most number of times with other T cells, which implies that NK cells may play an significant role in thyroid cancer. Selected signaling pathways with the greatest differences in communication probability were visualized (Figure 4D), providing a clearer understanding of how specific signaling programs are reprogrammed in tumor versus normal tissue contexts.

Collectively, this comparative analysis based on merged data provides a more statistically robust and biologically interpretable framework for understanding context-specific intercellular communication in thyroid cancer.

### 3.4. Treg Cell-Mediated Immune Communication Sheds Light on the Tumor Microenvironment: Insights from Cell Interactions

We initiated our analysis by comparing the conjectured interactions within the immune microenvironment of tumor and healthy tissues. We employed SingleCellSignalR, an algorithm adept in deducing intercellular networks from individual cell transcriptome data that operates on a manually curated ligand–receptor (LR) database [28].

SingleCellSignalR was applied to each patient’s data independently, emphasizing the reciprocal interactions between CD8^+^ T cells, Treg cells, CD4^+^ T cells, and NK cells. Our results revealed numerous predicted ligand–receptor interactions being abundantly present in both tumor and normal tissues (Figure 5A,B). Furthermore, antigen-presenting molecules and immune molecular interactions, such as those between HLA-A, HLA-B, HLA-C, HLA-E, and HLA-F with CD8A and CD8B, appeared to be significantly up-regulated in tumor tissues, while their presence in normal tissues was comparatively diminished. Additionally, observations indicated that antigen-presenting and immune molecular interactions were primarily enacted by Treg cells, NK cells, and CD4^+^ T cells exerting influence on CD8^+^ T cells, with negligible alterations in the intensity of LR mechanisms among other cell types. The scores of immunomodulatory interactions manifested a downgrade in tumor tissues, largely influenced by CLEC2D, CLEC2C, CLEC2B, and KLRB1. This indicates a less stimulated immune environment, ensuing difficulties in attracting peripheral immune cells towards the tumor.

Following this, we conceived an extensive study distinguishing the cellular interactions between the immune environments of tumor and normal tissues utilizing SingleCellSignalR. Initially, we enumerated the aggregate number of surmised ligand–receptor (LR) interactions for each T-cell category in all sample pairs (Figure 5C,D), uncovering numerous predicted interaction patterns entailing CD8^+^ T cells, Treg cells, NK cells, and CD4^+^ T cells in tumor tissues. Interestingly, interactions within CD8^+^ T cells were also observed. Conventionally, Treg cells, NK cells, and CD4^+^ T cells demonstrated fewer interactions, likely due to their overall lower gene expression profiles (Figure 5C). In contrast to tumor tissues, CD8^+^ T cells in normal tissues demonstrated less interactions alongside Treg cells, while Treg cells and CD4^+^ T cells virtually lacked interactions, albeit exhibiting significantly enhanced effects on NK cells (Figure 5D).

Seeking improved biological interpretability, we executed analyses of efferent and afferent signaling patterns in both tumor and normal tissues. Subsequently noted was that CD8^+^ T cells, Treg cells, NK cells, and CD4^+^ T cells in both tissue types established several efferent (ligand-expressing cells) and afferent (receptor-expressing cells) interactions (Figure 5E,F and Appendix A). Impressively, CD4^+^ T cells showed an elevated expression of efferent interactions; conversely, their afferent interaction expression was nearly void. Upon comparing tumor and normal tissues, it was observed that the signal intensity of MHC-2 in Treg cells and CD99 in CD8^+^ T cells were both bolstered in tumor tissues. Additionally, the signal intensity of LCK was higher in tumor tissues, while that of CLEC was found to be more accentuated in normal tissues. These data further support the hypothesis of thyroid cancer patients having an immunocompromised condition.

### 3.5. Development of a GNN Model Based on T-Cells’ Differential Genes for Risk Stratification and Prognosis Prediction

In order to identify genes with significant differences in expression within the entire T-cell population, we used significant differences expression genes of T cells in tumor tissues and normal tissues as features for subsequent modeling. The model was the proposed MLA-GNN, and the structure of the model is displayed in Figure 6A. Additionally, a differential gene analysis was performed on THCA data. Furthermore, WGCNA was employed to identify T-cell-associated genomes. Finally, modeling was carried out using the GNN model.

To examine whether or not the diagnostic model genes reflect functional alterations observed in specific T-cell subsets, we visualized their expression across tumor-infiltrating immune cells. Several marker genes used in the model—including PDCD1, TIGIT, LAG3, and IL2RB—were preferentially enriched in CD8^+^ T cells and Treg cells. These genes are well-established markers of immune suppression and T-cell dysfunction, aligning with our earlier findings that tumor tissues exhibit increased expression of inhibitory pathways and altered cell–cell communication (Appendix A).

Subsequently, the TCGA data were divided into training and validation sets in a 8:2 ratio. Based on the mid-optimal cut-off value, patients were classified into high-risk and low-risk groups. Survival analysis conducted on the training and validation sets demonstrated a significant disparity in the survival rate of THCA patients within the TCGA dataset. The Kaplan–Meier curve exhibited an unfavorable prognosis for patients in the high-risk group (*p* < 0.001), validating the model’s ability to stratify risk subtypes in the TCGA cohort (Figure 6B). The AUC of the ROC curve was used based on survival at 1, 3, and 5 years for predicting patient prognosis and was maintained at around 0.95 in the training set (Figure 6C). In addition, the risk score was found to help refine the prognostic stratification in the validation group, with high-risk patients having a significantly lower disease-free survival rate than low-risk patients (*p* = 0.0029) (Figure 6D). The ROC curve illustrated the good performance it achieved, with a 12-month AUC of 0.8 (Figure 6E).

### 3.6. Differential Analysis of T-Cell Infiltration in High-Risk and Low-Risk Patient Groups

In order to clarify the differences in the immune microenvironment between high-risk and low-risk groups of patients, we annotated the levels of CD8^+^ T cells, Treg cells, CD4^+^ T cells, and NKT cells using CIBERSORT, after applying batch correction to the gene expression matrix to reduce technical variability. The results demonstrated that the levels of infiltration by Treg cells significantly increased in the high-risk group (*p* < 0.001) (Figure 7A), and the levels of infiltration by CD4^+^ T cells were also notably increased in the high-risk group (*p* < 0.001) (Figure 7B). Conversely, the infiltration level of CD8^+^ T cells climbed markedly in the low-risk group (*p* = 0.035) (Figure 7C), while NKT cells also showed a significant increase in their level of infiltration in the low-risk group (*p* = 0.001) (Figure 7D). These results underscore that a high-risk group of patients is associated with an immunosuppressive microenvironment. These findings suggest a correlation between the immune microenvironment and patient risks. High-risk patients may have increased Treg and CD4^+^ T cells, known for their immunosuppressive roles. Low-risk patients may show a rise in CD8^+^ T cells and NKT cells, implying an effective anti-tumor response. This calls for more research to develop personalized immunotherapies and potential biomarkers for patient risk stratification.

### 3.7. Differential Gene Expression and Pathway Analysis in Low-Risk Versus High-Risk Groups: A Comparative Study Involving Gene Ontology and KEGG Analyses

In comparison to patients in the high-risk group, 166 genes were upregulated in the low-risk group, and as compared to patients in the low-risk group, 17 genes were found upregulated in the high-risk group (Appendix A). These results are demonstrated via a heat map representation.

A subsequent gene ontology (GO) analysis was conducted. This analysis identified that within the cellular component (CC) classification, the low-risk group exhibited maximum expression for the “antigen processing and presentation of exogenous peptide antigen via MHC class I” and “antigen processing and presentation of peptide antigen via MHC class II” pathways. Under the biological processes (BPs), the highest expression pathways were “collagen-containing extracellular matrix” and “external side of plasma membrane”. Moreover, the molecular function (MF) showed the highest expression for “MHC class II receptor activity” and “heparin binding” pathways (*p* < 0.05) (Appendix A).

In comparison, for the high-risk group within the CC category, the “apical plasma membrane” and the “vesicle lumen” pathways showed maximum expression. No obvious enriched pathways were identified for the BP category. Within the MF, the “integrin binding” and “extracellular matrix structural constituent” pathways showed the highest expression (*p* < 0.05) (Appendix A).

A subsequent KEGG analysis found that, in the low-risk group, the MAPK signaling pathway and thyroid hormone synthesis were highly expressed (*p* < 0.05) (Appendix A). However, in the high-risk group, the “Glycosphingolipid biosynthesis − lacto and neolacto series” pathway was highly expressed (*p* > 0.05) (Appendix A).

### 3.8. Differential Mutational Profiles and Tumor Mutational Burden in High-Risk and Low-Risk Groups: BRAF Mutations

In comparison to patients in the low-risk group, the high-risk group exhibited a higher mutation rate, accounting for 78.95% and 68.39%, respectively. BRAF was identified as the most prevalent mutated gene, which are closely associated with cell proliferation. In the high-risk group, BRAF mutations were observed in 68% of patients. In contrast, the low-risk group showed BRAF mutations in approximately 51% of patients (Figure 8A).

Furthermore, we calculated the difference in BRAF mutation and non-BRAF mutation between high-risk and low-risk groups (Figure 8B). The result showed that the proportion of BRAF mutations in the high-risk group was higher than in the low-risk group (*p* = 0.002).

Additionally, we analyzed the expression of tumor mutational burden (TMB) in the high-risk and low-risk groups. The results demonstrated a significantly higher TMB expression in the high-risk group compared to the low-risk group (*p* < 0.001). This indicates a greater number of mutational events in the tumor cells of high-risk group patients, which may contribute to tumor instability and progression (Figure 8C).

In summary, there are notable differences in mutational profiles between the high-risk and low-risk groups. The high-risk group exhibited a higher mutation rate, with BRAF being the major mutated gene. This indicates that BRAF mutations may contribute to poor prognosis in thyroid cancer patients. Additionally, the high-risk group displayed elevated TMB levels, which may be associated with the biological characteristics of the tumor and patient prognosis.

### 3.9. Subgroup Analyses Reveal Improved Disease-Free Survival in Low-Risk Group: Age, Gender, and Stage-Specific Impact in Patients

After conducting subgroup analysis based on age and gender within the high-risk and low-risk groups, we observed significant differences in DFS between these subgroups. Specifically, among patients aged above 55 and below 55, those in the low-risk group exhibited significantly longer DFS compared to the high-risk group (Appendix A) (*p* < 0.0001). Additionally, in both male and female patients, the low-risk group demonstrated significantly improved DFS compared to the high-risk group (Appendix A) (*p* < 0.0001), with all ROC curve scores exceeding 0.95 (Appendix A).

Furthermore, we performed stage-specific analysis within the high-risk and low-risk groups. The results revealed that, in stage 1, 2, 3, and 4 patients, those in the low-risk group had significantly longer DFS than those in the high-risk group (Appendix A) (*p* < 0.0001), with ROC curve scores exceeding 0.95 (Appendix A).

Moreover, we conducted subgroup analysis based on T stage and N stage within the high-risk and low-risk groups. Consistently, irrespective of T1, T2, T3, T4 stages, or N1, N2, N3, N4 stages, patients in the low-risk group had significantly longer DFS compared to those in the high-risk group (Appendix A) (*p* < 0.0001), with ROC curve scores also exceeding 0.95 (Appendix A).

To summarize, our findings indicate that, within age and gender subgroups, as well as stage subgroups, patients in the low-risk group showed significantly improved DFS compared to those in the high-risk group. These results suggest that these subgroups may serve as important prognostic indicators for predicting disease-free survival in patients.

## 4. Discussion

This study constructed a thyroid cancer prognostic model using integrated RNA-seq and scRNA-seq data, identifying T-cell-based differentially expressed genes. A risk model was developed and validated, revealing a significant disease-free survival difference between high- and low-risk groups. Patients with high-risk scores had higher disease recurrence rates and tumor mutational burden, implying a potential favorable response to immunotherapy. These findings may guide future treatment strategies.

The current study delves into an advanced understanding of the thyroid cancer microenvironment through an integrated analysis of scRNA-seq data and bulk RNA sequencing data. This data integration allows for an in-depth single-cell resolution understanding of the cellular composition and heterogeneity within thyroid tumors and adjacent tissues, thereby spotlighting the potential role of specific cellular subpopulations in tumorigenesis. Our scRNA-seq analysis delineated the multifaceted cell types present within the thyroid tumor microenvironment. The heterogeneity observed provides evidence for the complexity of the tumor and its surrounding tissues. This is in line with the results of a recent study by Pu et al., which highlights the tumor microenvironment heterogeneity in thyroid cancer [29]. The role of cellular diversity within tumors has been shown to impact the disease progression, therapeutic responses, and overall survival rates in patients with various cancers [30].

Of paramount importance is the revelation of a pronounced difference in the proportion of T-cell subtypes within tumor tissue compared to normal tissue. Our findings emphasize T cells’ crucial role in the tumor microenvironment, underlining their prognostic potential for thyroid cancer. An extensive comparative analysis of T-cell subpopulations brought to light distinct metabolic activities, specifically, the enrichment of glycolytic pathways in Treg cells present in the tumor tissues. This finding is a significant extension of the study by De Rosa et al. [31], which established a direct relationship between metabolic reprogramming in T cells and their influence on the tumor microenvironment. This shift in metabolic pathways could potentially impact the phenotype and function of T cells, thereby affecting the tumor microenvironment. This indicates that Tregs could act as a promising biomarker for the prognosis and progression of the disease. Building upon these insights, the SingleCell and SignalR tools were employed to uncover distinct patterns of intercellular communication within the tumor immune microenvironment. The results from our study resonate with the findings of Luo et al. [32]: HLA and CD8 are expressed at higher levels between Treg cells and CD8^+^ T cells in tumor tissues. These interactions, especially those involving antigen-presenting and immune molecular interactions, were found to be particularly pronounced in tumor tissues, emphasizing their significant role in the tumor–immune interplay. The link between glycolytic pathways in Tregs and immune suppression has been highlighted in previous studies, but further mechanistic insights are needed. Our study suggests that Treg cells enhance their glycolytic activity in the tumor microenvironment. This metabolic shift facilitates lactate secretion, which in turn inhibits CD8^+^ T-cell function, impairing their anti-tumor response. This mechanism contributes to the immunosuppressive environment within tumors and underscores the potential of targeting Treg cell metabolism as a therapeutic strategy in cancer immunotherapy [33].

In light of the observed metabolic reprogramming in tumor-infiltrating Treg cells—particularly their elevated glycolytic activity—these features may offer translational value as novel biomarkers. For instance, glycolysis-related genes expressed by Treg cells (such as PKM and LDHA) could potentially be detected in peripheral blood via liquid biopsy, using cell-free RNA or exosomal RNA profiling. Such an approach may enable the non-invasive assessment of tumor immune status, prognostic risk stratification, or even the early monitoring of immunotherapy response in thyroid cancer patients [34,35].

The MLA-GNN model was constructed for risk stratification and prognosis prediction. The model stratified thyroid cancer patients into high-risk and low-risk groups, demonstrating significant differences in DFS. A similar model was used by Zhang et al. [36], showing the potential of such machine learning models in risk stratification and prognosis prediction in thyroid cancer patients. Significant differences were observed in immune cell infiltration levels between the high-risk and low-risk groups. Recent studies have shown a correlation between immune cell infiltration levels and patient prognosis, suggesting that an immunosuppressive microenvironment could be associated with a higher risk of disease progression [37,38]. This could serve as a crucial indicator for the immune response in thyroid cancer and may pave the way for potential biomarker discovery and the development of personalized immunotherapies. Furthermore, differential mutational profiles were noted between the high-risk and low-risk groups. Patients with high-risk scores had higher tumor mutational burden. While elevated TMB has been associated with immunotherapy response in certain cancers, its predictive value in thyroid cancer remains uncertain. In accordance with recent studies, BRAF and NRAS mutations were predominantly observed in the high-risk group. This association aligns with their known role in cell proliferation and thyroid cancer progression, as well as the prognostic value of RAS mutations in thyroid cancer [39]. However, recent studies suggest that features of the immune microenvironment—such as T-cell infiltration and immune activation—may be more relevant predictors of immunotherapy benefit in thyroid cancer. CD8^+^ T cells and the expression of immune checkpoint molecules may better reflect the immunogenic potential of thyroid tumors. Therefore, future clinical strategies may benefit from integrating TMB with immune profiling to identify patients likely to respond to immunotherapy [40]. Although this study primarily focused on transcriptomic-level distinctions among thyroid cancer subgroups, it is worth noting that oxidative stress and redox imbalance have been increasingly recognized as drivers of tumor progression, immune modulation, and metabolic rewiring in thyroid malignancies. These processes may underlie some of the differential gene expression patterns captured by MLA-GNN. For instance, redox signaling can regulate T-cell function, apoptosis resistance, and mitochondrial metabolism, all of which are pathways reflected in some of the subtype-specific marker genes we identified. While speculative, the incorporation of oxidative stress–related transcriptomic signatures in future studies may offer further insight into the biological underpinnings of MLA-GNN-based stratification [41].

Several limitations of this study should be acknowledged. First, although our model demonstrated high prognostic performance in internal validation within the TCGA cohort, no large-scale external dataset was available for independent validation. The GEO cohort used in this study served only for single-cell characterization and is too small to support claims of model generalizability. Second, although our model demonstrated robust predictive performance, its clinical interpretability is currently limited by the available clinical annotations in the TCGA dataset. Future work leveraging richer clinical datasets or prospective cohorts will be necessary to validate the associations between model-identified markers and specific clinical phenotypes, thus enhancing the translational applicability of the model. Lastly, the exceptionally high AUC values observed in subgroup analyses may reflect potential overfitting, especially in subsets with limited sample sizes. Prospective validation in multi-center cohorts is warranted to evaluate the real-world performance and clinical utility of our proposed risk model.

In conclusion, this study emphasizes the potential of scRNA-seq technology to illuminate the intricate cellular landscape of thyroid cancer and the consequential role of T cells within the tumor microenvironment. Our results reinforce the potential of the MLA-GNN model in prognosis prediction and patient stratification into high-risk and low-risk groups. These findings could lay the groundwork for future research in developing personalized therapeutic strategies and potential prognostic biomarkers for thyroid cancer.

## 5. Conclusions

In summary, this study comprehensively integrates bulk RNA-seq and single-cell RNA-seq data to construct a T-cell–related prognostic model for thyroid cancer using a multi-layer attention graph neural network (MLA-GNN). The model demonstrates robust and stable predictive power for disease-free survival (DFS) across subgroups and highlights the critical involvement of T-cell subpopulations—particularly regulatory and exhausted CD8^+^ T cells—in shaping the immunosuppressive tumor microenvironment. Importantly, the identified risk-associated genes are not only statistically significant but also biologically relevant, reflecting functional dysregulation within key T-cell compartments. These findings deepen our understanding of immune-mediated mechanisms in thyroid cancer and suggest that T-cell–based molecular signatures may serve as promising biomarkers for prognosis assessment and immunotherapy response prediction. By incorporating immune profiling into prognostic modeling, this work provides a foundation for more precise patient stratification and opens new avenues for developing individualized treatment strategies in thyroid cancer.

## Figures and Tables

**Figure 1 cancers-17-02411-f001:**
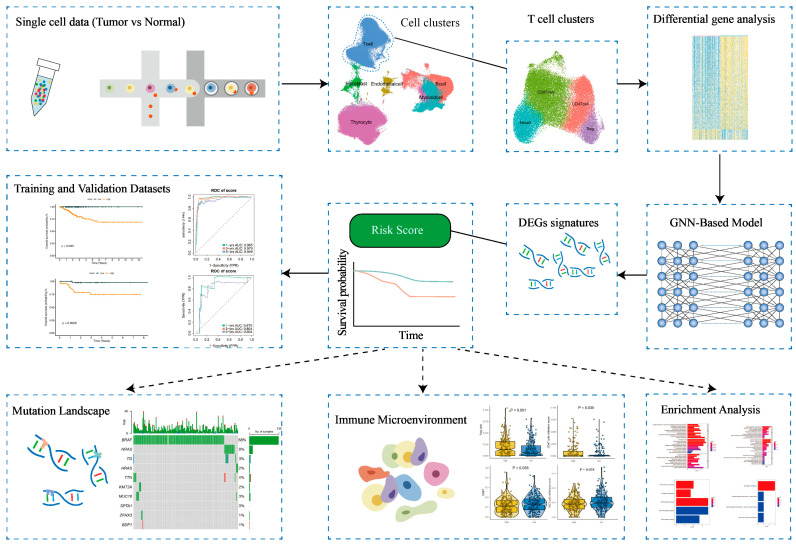
Workflow diagram of the study.

**Figure 2 cancers-17-02411-f002:**
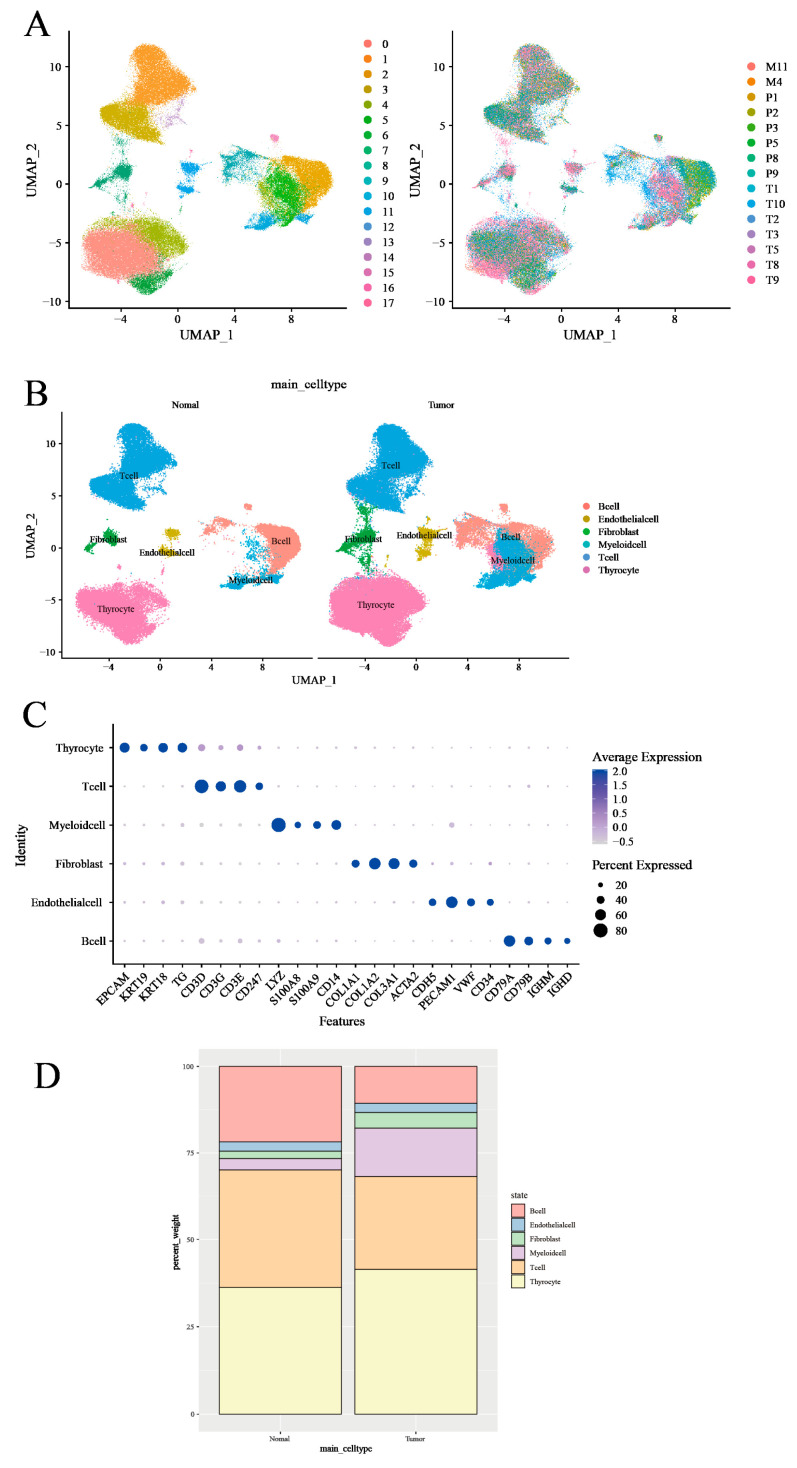
Transcriptomic analysis of immune environments in thyroid cancer. (**A**) UMAP plot of scRNA-seq data from all 158,557 cells colored by patient. (**B**) UMAP plot of scRNA-seq data colored by cell type in tumor tissues and normal tissues. (**C**) DotPlot was generated to visualize the expression level of main cell type markers in the specified cell subsets. (**D**) Proportion (% of total cells) of main cell types in tumor tissues and normal tissues.

**Figure 3 cancers-17-02411-f003:**
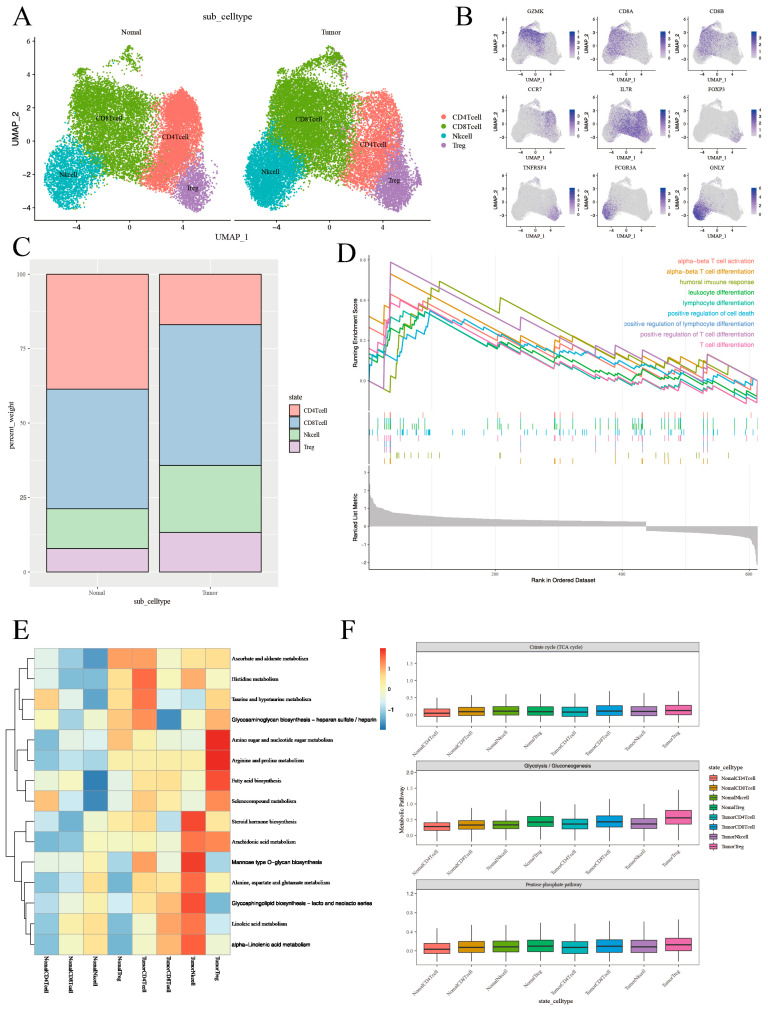
The T-cell phenotypic landscape and related pathways of tumor tissues and normal tissues in thyroid cancer. (**A**) UMAP plot of scRNA-seq data from 36,443 T cells colored by Seurat cluster, annotated with the indicated cell type labels. (**B**) UMAP plot of T-cell markers in the specified cell subsets. (**C**) Proportion (% of T cells) of T-cell types in tumor tissues and normal tissues. (**D**) GSEA analysis showing T-cell-related signaling pathways in tumor tissues and normal tissues. (**E**) Heatmap showing differences in metabolic pathway activity (scored per cell by GSVA) in 4 T-cell subclusters. The full list of metabolic pathways is in Appendix A. (**F**) Boxplots showing the expression level of genes from Citrate cycle (TCA cycle) and Glycolysis/Gluconeogenesis, as well as the Pentose phosphate pathway in T-cell subclusters in tumor tissues and in normal tissues. Each boxplots represents the score of each T-cell subcluster.

**Figure 4 cancers-17-02411-f004:**
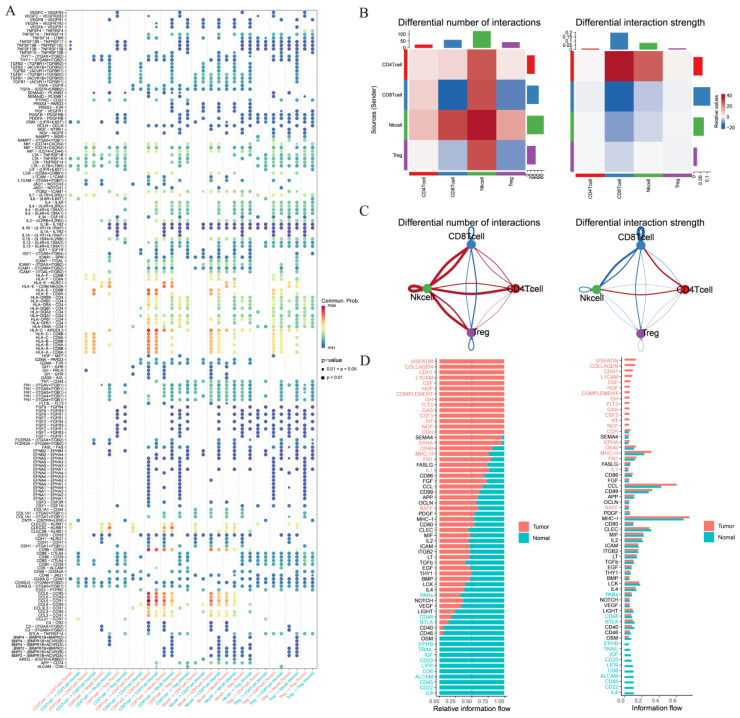
Ligand–receptor analysis predicts different T-cell subclusters’ crosstalk in thyroid cancer. (**A**) Differential number of intercellular interactions between cell subtypes in thyroid cancer. (**B**) Differential interaction strength among immune cell types in thyroid cancer. (**C**) Relative information flow of signaling pathways in thyroid cancer. (**D**) Selected signaling pathways exhibiting notable differences in relative communication probability. This comparative analysis was performed on the merged dataset using CellChat.

**Figure 5 cancers-17-02411-f005:**
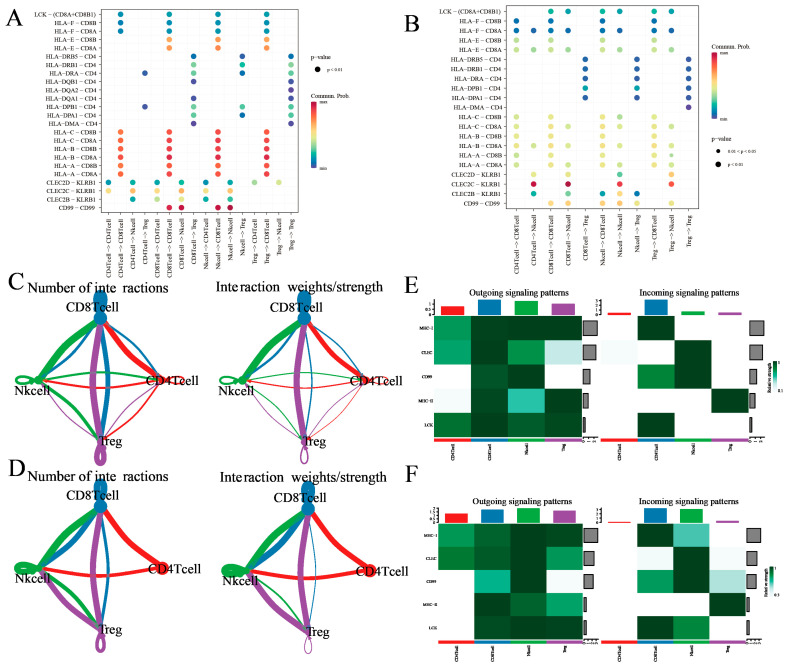
Ligand–receptor analysis predicts different T-cell subclusters’ crosstalk between tumor and normal tissues. (**A**,**B**) Enrichment of selected ligand–receptor interactions in either tumor tissues or normal tissues for the given cell type pairs. selections were made by considering evidence from existing literature and ensuring biological interpretability. White squares denote interactions with an enrichment *p* value > 0.05. Two-sided Wilcoxon rank sum test was used for statistical analysis. (**C**,**D**) Social graph depicting the interaction weights/strength and the number of interactions between the 4 T-cell types in tumor tissues and normal tissues. (**E**,**F**) Heatmap showing the expression of ligand–receptor interactions of the 4 T-cell types in outgoing signaling patterns and incoming signaling patterns in tumor tissues and normal tissues.

**Figure 6 cancers-17-02411-f006:**
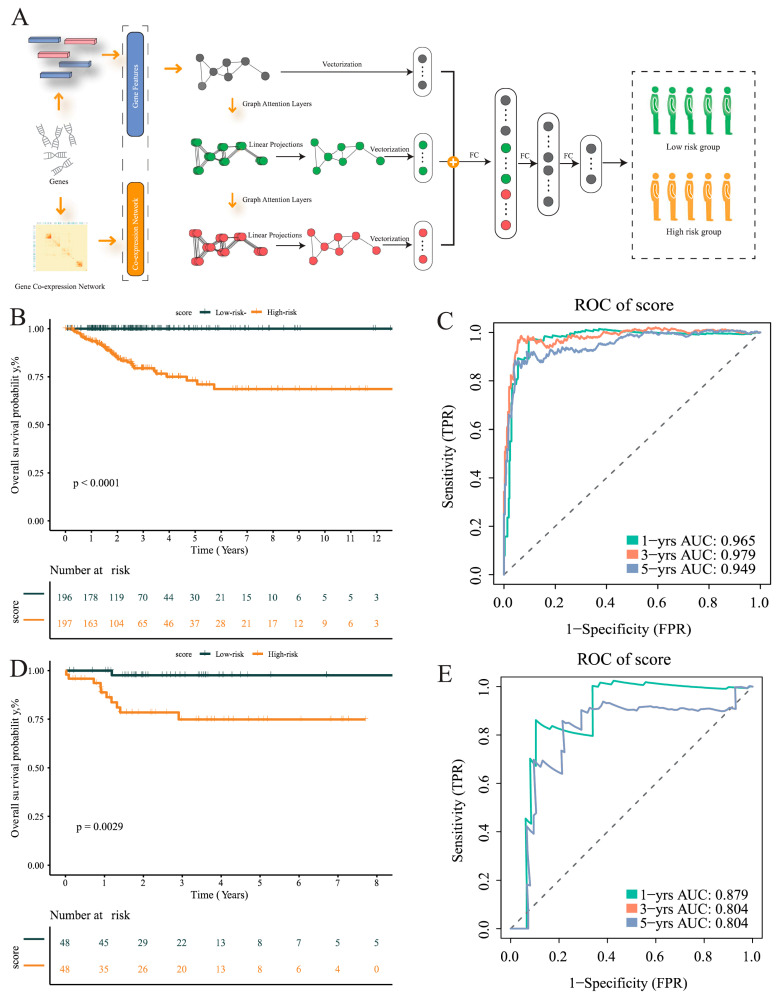
Construction of multi-level attention graph neural network (MLA-GNN) in the TCGA cohort. (**A**) Overview of the proposed multi-level attention graph neural network (MLA-GNN). (**B**) KM curves of the validation group. (**C**) ROC curves for the validation group. (**D**) KM curves for the training group. (**E**) ROC curves for the training group.

**Figure 7 cancers-17-02411-f007:**
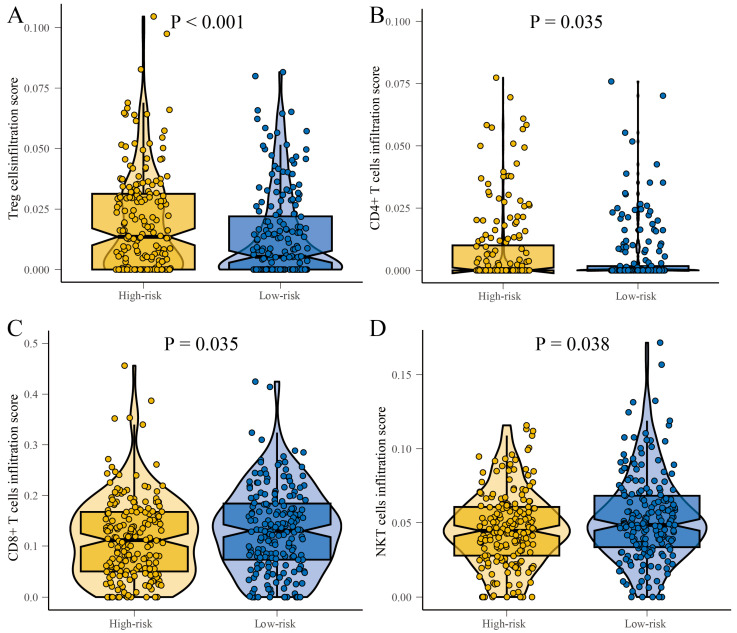
Boxplots showing the differences in immune infiltration of T cells in the high-risk and low-risk groups. (**A**) Treg cells. (**B**) CD4^+^ T cells. (**C**) CD8^+^ T cells. (**D**) NKT cells.

**Figure 8 cancers-17-02411-f008:**
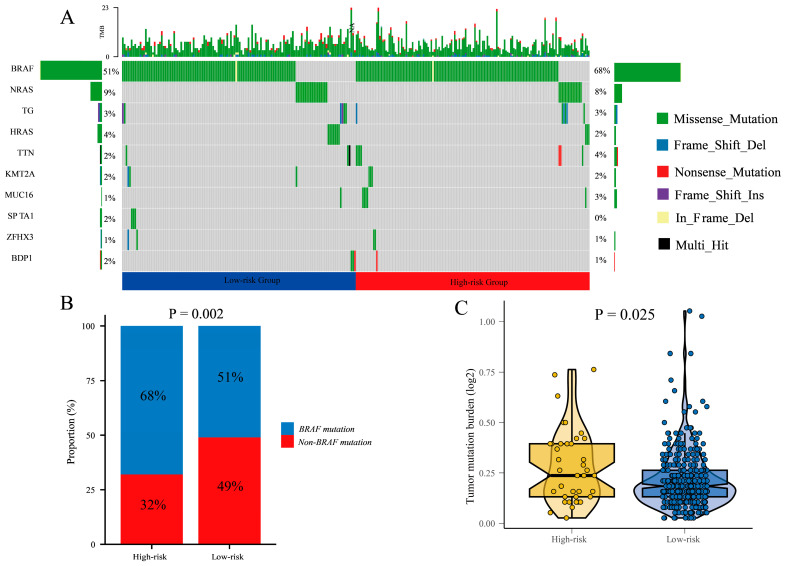
The mutation landscape in the high-risk and low-risk groups. (**A**) the mutation landscape of the top 10 genes with mutation frequency in differential risk subgroups. (**B**) Description of the statistical measurement mutation details of BRAF, which is the most common mutation type. (**C**) Comparison of tumor mutation burden (TMB) between different risk groups.

**Table 1 cancers-17-02411-t001:** Clinical information of patients with thyroid cancer in this study.

Corhort	TCGA-THCA	GSE184362
Number of patients	n = 489	n = 11
Age (Mean ± SD)	46.53 ± 15.35	-
Follow up time (Mean ± SD) (years)	3.11 ± 2.65	-
Follow up status		
Alive	442 (90.4%)	-
Dead	47 (9.6%)	-
Gender		
Male	130 (26.6%)	-
Female	359 (73.4%)	-
Clinical stage		
Stage I	281 (57.5%)	-
Stage II	50 (10.2%)	-
Stage III	105 (21.5%)	-
Stage IV	51 (10.4%)	-
Unknown	2 (0.4%)	-
T stage		
T0	-	4 (36.4%)
T1	140 (28.6%)	2 (18.2%)
T2	163 (33.3%)	-
T3	166 (34.0%)	-
T4	18 (3.7%)	5 (45.4%)
Unknown	2 (0.4%)	-
M stage		
M0	273 (55.9%)	7 (63.6%)
M1	8 (1.6%)	4 (36.4%)
Unknown	208 (42.5%)	-
N stage		
N0	225 (46.0%)	2 (18.2%)
N1	217 (44.4%)	9 (81.8%)
Unknown	47 (9.6%)	-
Histologic subtype		
Classical	-	8 (72.7%)
Follicular variant	-	2 (18.2%)
Tall cell variant	-	1 (9.1%)

## Data Availability

All data was obtained from The Cancer Genome Atlas (TCGA, https://tcga-data.nci.nih.gov/tcga/) and Gene Expression Omnibus (GEO, https://www.ncbi.nlm.nih.gov/geo/). URL (accessed on 5 May 2024). For data resources, please contact the corresponding author. To ensure the reproducibility of our research, the code used in this study is available upon request from the corresponding author, Miaoyun Long. We encourage other researchers to reach out for any additional resources or clarifications regarding our methods.

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
