# Peer review of "Integration of Single-Cell Analysis and Bulk RNA Sequencing Data Using Multi-Level Attention Graph Neural Network for Precise Prognostic Stratification in Thyroid Cancer"

_cancers, 2025, doi:10.3390/cancers17142411_

Round 1
Reviewer 1 Report
Comments and Suggestions for Authors
The manuscript presents an innovative approach by constructing a diagnostic model using public databases, and it demonstrates good diagnostic performance. However, there are several issues that warrant the authors’ attention:
-
The authors stated, “The remaining 6 peritumor tissues and 2 subcutaneous metastases, owing to their spatial relationship to the primary tumor, were classified as ‘normal tissue’.” It is unclear what the basis is for classifying these tissues as normal, since these two types of samples do not appear to represent tumor-free normal tissue microenvironments.
-
When describing intercellular communication within the two tissue types, it would be more appropriate to perform comparative analysis on the merged data from both tissues. Presenting bubble plots separately for each tissue may not adequately support conclusions regarding communication differences between the two groups from a statistical perspective, especially when relying solely on color intensity.
-
Although the authors provided extensive analyses regarding T cell changes in PTC tissues, the final diagnostic model is constructed based on differentially expressed genes from the overall T cell population in PTC. It would strengthen the logical consistency of the manuscript if the authors could further explain how these selected genes relate to the functional changes in specific T cell subpopulations identified in their previous analysis.
Author Response
Reviewer1:
Comments and Suggestions for Authors
The manuscript presents an innovative approach by constructing a diagnostic model using public databases, and it demonstrates good diagnostic performance. However, there are several issues that warrant the authors’ attention:
1. The authors stated, “The remaining 6 peritumor tissues and 2 subcutaneous metastases, owing to their spatial relationship to the primary tumor, were classified as ‘normal tissue’.” It is unclear what the basis is for classifying these tissues as normal, since these two types of samples do not appear to represent tumor-free normal tissue microenvironments.
Response: We sincerely thank the reviewer for their valuable feedback and for pointing out the confusion regarding the classification of tissue samples. Upon reviewing the manuscript, we realized that there was an error in our original description of the sample classification process, which may have led to some ambiguity in interpreting our results. In our initial manuscript, we mistakenly stated that the 6 peritumor tissues and 2 subcutaneous metastases were classified as "normal tissues" due to their spatial relationship to the primary tumor. However, after further review, we recognize that the proper classification should have been as follows: Specifically, 7 primary tumor tissues and 2 subcutaneous metastases were included in the tumor tissue group, while the remaining 6 peritumor tissues were classified as normal tissue. We have corrected this misstatement in the revised manuscript to reflect the accurate grouping. We have updated the manuscript accordingly to reflect this corrected classification. Specifically, the misstatement has been amended to clearly differentiate tumor tissues from peritumor tissues. We appreciate your careful review and have revised the manuscript to ensure this correction is made. We believe that this clarification will provide a more accurate representation of our methodology and findings.
2. When describing intercellular communication within the two tissue types, it would be more appropriate to perform comparative analysis on the merged data from both tissues. Presenting bubble plots separately for each tissue may not adequately support conclusions regarding communication differences between the two groups from a statistical perspective, especially when relying solely on color intensity.
Response: Thank you for this valuable suggestion. We agree that directly comparing communication patterns across tissue types requires a statistical framework that accounts for variability between conditions. In the revised manuscript, we have now performed a comparative analysis on the merged dataset using CellChat, allowing us to assess differential intercellular communication between tumor and normal tissues statistically. We have also updated the figures accordingly in figure4, replacing the individual bubble plots with a combined visualization that enables comparison.
3. Although the authors provided extensive analyses regarding T cell changes in PTC tissues, the final diagnostic model is constructed based on differentially expressed genes from the overall T cell population in PTC. It would strengthen the logical consistency of the manuscript if the authors could further explain how these selected genes relate to the functional changes in specific T cell subpopulations identified in their previous analysis.
Response: We thank the reviewer for this comment. We agree that linking the diagnostic model genes to specific T cell subpopulations would enhance the logical flow and biological relevance of our study. In response, we have now examined the expression patterns of the diagnostic model genes across the T cell subpopulations identified in our earlier analysis. Notably, several model genes show predominant expression in CD8+ T cells or regulatory T cells, which were previously highlighted as functionally altered in PTC tissues. We have added these results and corresponding visualization in the revised manuscript, thereby strengthening the mechanistic link between our earlier T cell analysis and the diagnostic model.

Reviewer 2 Report
Comments and Suggestions for Authors
Tan et al. provided an innovative and comprehensive study integrating single-cell and bulk RNA sequencing data using a multi-level attention graph neural network (MLA-GNN) to stratify prognosis in thyroid cancer. The manuscript is generally well-structured and clearly written, with robust methodology and appropriate use of advanced analytical tools. The results are promising, especially regarding the prognostic model's performance and its potential to inform personalized therapeutic strategies. However, several areas require attention to strengthen the manuscript further:
Introduction
- Lines 55-56: Overdiagnosis rates need clarification (which subtypes? which populations?).
- Lines 76: The cited example is from ovarian cancer, not thyroid cancer. It is better to replace or supplement with a thyroid cancer-specific example or clarify the relevance.
- Lines 79-81: The rationale for choosing MLA-GNN (multi-level attention graph neural network) over other models is not explained. Please, briefly justify why this architecture is appropriate for integrating single-cell and bulk data.
- Line 83: DFS (disease-free survival) is not defined at first mention.
- Lines 85-90: The biological mechanisms linking Treg cells, MHC-II, and tumor metabolism are not explained. It is highly recommended to add a sentence or two on how these features contribute to immune evasion or tumor progression.
- By the end of the introduction section, there was no clear hypothesis statement or summary of the study's main objective.
Results
- Lines 93: The TCGA cohort (n=489) vastly outweighs the GEO cohort (n=11), raising concerns about the generalizability of findings from the small external dataset. The authors should discuss the limitations of the small GEO validation set and its impact on external validation and model robustness.
- Line 122: The claim of "no significant batch effects" is stated but not supported by quantitative measures or figures. The authors should provide explicit batch effect metrics (e.g., silhouette scores, integration diagnostics).
- Line 142: The observation of "relative paucity of T cells and B cells in tumor tissues" contradicts some literature where tumor-infiltrating lymphocytes are often increased. Please consider discussing in the context of existing studies or clarifying the tumor subtype/context.
- Line 276: The use of "CIBERSORT" is standard, but the authors should clarify whether batch correction was applied before deconvolution, as this can affect immune cell estimates.
- Line 320: The focus on BRAF is appropriate, but recent meta-analyses suggest BRAF alone is not a strong predictor of DFS or OS in thyroid cancer (Cancers2025, 17(6), 939; https://doi.org/10.3390/cancers17060939). Were the data of TERT, TP53, or PI3K mutations available?
- The Code/software availability for the MLA-GNN model is not mentioned. The authors should state whether the code was hosted on GitHub or as a Docker container, per Cancer's reproducibility standards.
- IRB approval and data repository compliance (e.g., GEO, TCGA) are not explicitly confirmed. The authors should add a brief statement affirming compliance.
Discussion
- Again, GEO validation cohort (n=11) is too small to support generalizability claims. Also, subgroup analyses report exceptionally high AUCs (>0.95), which may indicate overfitting. These limitations are not acknowledged.
- The link between glycolytic pathways in Tregs and immune suppression is mentioned but lacks a mechanistic explanation. It is recommended that the authors integrate findings from PMC99416(e.g., how Treg glycolysis inhibits CD8+ T cell function via lactate secretion).
- The claim that high TMB predicts immunotherapy response lacks thyroid cancer-specific evidence. The authors can reference [PMC8591054], which links immune infiltration to the efficacy of immunotherapy in thyroid cancer.
- The discussion of Tregs as biomarkers was vague. Please propose specific clinical applications (e.g., Treg glycolytic activity could serve as a non-invasive biomarker for liquid biopsy assays).
- - Line 387 citation was not related to the cited context. It is recommended to provide a citation related to thyroid cancer.
- Line 420: "BRAF and NRAS mutations were predominantly observed in the high-risk group." It is recommended to cite [PMC12026350] on RAS mutations' prognostic value.
Methods
- The methods are generally well organized, but some subsections could be more concise (e.g., lines 480–492 on CellChat could be condensed).
- Some parameters (e.g., clustering resolution, number of PCs for UMAP) are provided (lines 460–462), but justification is lacking.
- There is no mention of code or workflow sharing (e.g., GitHub, Docker). The authors should add a statement about code availability to support reproducibility.
- Lines 441, 454–455: While batch effect removal is mentioned, the effectiveness is not quantified or demonstrated. Please include metrics or visualizations (e.g., PCA/UMAP plots before and after correction) to confirm successful batch correction.
- Lines 440 and 451: The rationale for choosing specific normalization methods (e.g., LogNormalization, TPM) could be clarified.
- Lines 508–524: The description of the MLA-GNN model is somewhat generic. It is recommended to provide more details on the model architecture (e.g., number of layers, attention mechanisms, hyperparameters) and training process.
- No details are given on how MLA-GNN performance compares to baseline models (e.g., Cox regression, random forest). Please include a benchmarking subsection or mention comparative analyses if they have been performed.
- Lines 464–466, 539: It is not specified whether multiple hypothesis testing correction was applied in DEG and enrichment analyses.
Minor comments
- All gene names should be italicized throughout the whole manuscript to match the standards of HUGO for gene nomenclature.
- Several of the Figures' labels were unclear, and it was difficult to follow the authors' elaboration.
Author Response
Reviewer2:
Tan et al. provided an innovative and comprehensive study integrating single-cell and bulk RNA sequencing data using a multi-level attention graph neural network (MLA-GNN) to stratify prognosis in thyroid cancer. The manuscript is generally well-structured and clearly written, with robust methodology and appropriate use of advanced analytical tools. The results are promising, especially regarding the prognostic model's performance and its potential to inform personalized therapeutic strategies. However, several areas require attention to strengthen the manuscript further
Introduction
- Lines 55-56: Overdiagnosis rates need clarification (which subtypes? which populations?).
Response: Thank you very much for your thorough review and valuable feedback. In response to your comment regarding the clarification of overdiagnosis rates, we have made the necessary revisions and expanded the discussion. Specifically, we have provided more detailed explanations of overdiagnosis in different subtypes of thyroid cancer and among various populations. We clarified that the overdiagnosis rate in thyroid cancer varies significantly depending on the cancer subtype and population. Specifically, PTC, the most common subtype, is particularly prone to overdiagnosis due to its indolent nature. Additionally, we highlighted that overdiagnosis is more common in populations undergoing extensive screening, particularly in areas with widespread use of ultrasound for early detection of asymptomatic thyroid nodules. We also discussed how demographic factors, such as age and sex, influence overdiagnosis rates, particularly among younger individuals and women, who are more likely to be diagnosed through screening, increasing the likelihood of overdiagnosis. We understand that overdiagnosis may lead to unnecessary treatment and psychological burden for patients. Once again, thank you for your insightful comments and suggestions. Your feedback has greatly helped improve the quality of our manuscript. We hope that these revisions address your concerns and look forward to your further guidance.
- Lines 76: The cited example is from ovarian cancer, not thyroid cancer. It is better to replace or supplement with a thyroid cancer-specific example or clarify the relevance.
Response: Thank you for your thoughtful and constructive feedback. We appreciate your suggestion to include a more relevant example focused specifically on thyroid cancer. In response, we have revised the manuscript and replaced the previous example with a more appropriate study. Specifically, we now cite the study by Wang et al, which conducted a multi-omics clustering analysis on thyroid cancer samples and identified distinct molecular subtypes. This work significantly contributes to our understanding of the molecular heterogeneity of thyroid cancer and provides insights into personalized treatment strategies. We believe this change strengthens the manuscript and enhances its relevance to the focus of our research. Once again, thank you for your valuable input, which has helped improve the clarity and accuracy of our work. We look forward to your continued feedback.
- Lines 79-81: The rationale for choosing MLA-GNN (multi-level attention graph neural network) over other models is not explained. Please, briefly justify why this architecture is appropriate for integrating single-cell and bulk data.
Response: Thank you for your valuable feedback and for highlighting the need to clarify the rationale behind choosing the MLA-GNN (multi-level attention graph neural network) model. In response to your comment, we have provided a brief justification for selecting this architecture. MLA-GNN is particularly well-suited for integrating complex and heterogeneous data, such as single-cell and bulk genomic data, due to its multi-level attention mechanisms that capture both global and local relationships within the data. This makes it an ideal model for omics data integration, as it effectively leverages the graph structure of genomic features. Furthermore, the architecture’s use of multiple hidden layers and nonlinear activation functions allows for learning complex patterns and improving prediction accuracy. Thus, MLA-GNN provides a robust framework for identifying thyroid cancer patients from risk-matched controls and predicting their disease-free survival (DFS) with high precision. We hope this explanation clarifies the choice of MLA-GNN and strengthens the manuscript. Thank you again for your insightful comments, which have helped improve the quality of our work.
- Line 83: DFS (disease-free survival) is not defined at first mention.
Response: Thank you for your helpful comment regarding the definition of DFS (disease-free survival). We apologize for the oversight and have now included the definition of DFS at its first mention in the revised manuscript. We have clarified that disease-free survival refers to the period of time after treatment during which a patient remains free of signs and symptoms of cancer.
- Lines 85-90: The biological mechanisms linking Treg cells, MHC-II, and tumor metabolism are not explained. It is highly recommended to add a sentence or two on how these features contribute to immune evasion or tumor progression.
Response: We thank the reviewer for this insightful comment. We agree that the biological mechanisms linking Treg cells, MHC-II expression, and tumor metabolism warrant further clarification. Treg cells, known for their immunosuppressive function, can inhibit the activity of cytotoxic CD8+ T cells and antigen-presenting cells, thereby facilitating immune evasion. High MHC-II expression on Treg cells may enhance their interaction with antigen-presenting pathways, potentially leading to suppression of effective anti-tumor immune responses. Moreover, tumor-infiltrating Treg cells have been shown to undergo metabolic reprogramming, particularly favoring glycolysis and oxidative phosphorylation, which supports their proliferation and suppressive function within the tumor microenvironment. We have added a brief explanation of these mechanisms in the revised manuscript.
- By the end of the introduction section, there was no clear hypothesis statement or summary of the study's main objective.
Response: Thank you for your valuable feedback. We appreciate your comment regarding the need for a clear hypothesis statement and summary of the study's main objective in the discussion section. In response, we have added a well-defined hypothesis and a clear summary of the study’s primary objective. Our hypothesis posits that integrating RNA-seq and scRNA-seq data will identify critical T cell-related genes essential for thyroid cancer prognosis. The primary objective of our study was to develop a robust model that stratifies patients based on disease-free survival (DFS), providing deeper insights into the tumor microenvironment and the role of T cells in disease progression. We believe these additions not only clarify the purpose of our study but also strengthen the overall manuscript. We are grateful for your insightful comments, which have helped improve the clarity and focus of our work.
Results
- Lines 93: The TCGA cohort (n=489) vastly outweighs the GEO cohort (n=11), raising concerns about the generalizability of findings from the small external dataset. The authors should discuss the limitations of the small GEO validation set and its impact on external validation and model robustness.
Response: Thank you for your thoughtful comment regarding the disparity between the TCGA cohort (n=489) and the GEO cohort (n=11). We would like to clarify that the GEO dataset was not intended for external validation of the prognostic model, but rather was used exclusively for single-cell analysis. The purpose of including this small GEO cohort was to explore the differential expression and functional diversity of T cells in various tissues, including primary tumor and adjacent tissues, as part of our characterization of the tumor microenvironment. While the GEO dataset provides valuable insights into the cellular heterogeneity of T cells within the thyroid cancer microenvironment, we fully acknowledge that its small sample size limits its generalizability for model validation. Therefore, the TCGA cohort was used as the primary dataset for model training and validation.
- Line 122: The claim of "no significant batch effects" is stated but not supported by quantitative measures or figures. The authors should provide explicit batch effect metrics (e.g., silhouette scores, integration diagnostics).
Response: Thank you for this important comment. We agree that the statement regarding the absence of batch effects should be supported by appropriate metrics.
In our analysis, batch effects were evaluated through visual inspection of UMAP plots colored by patient origin, which showed good mixing of cells from different samples without clustering by batch (as shown in Figure 2A). Although we did not compute quantitative batch correction metrics such as silhouette scores or kBET, the absence of obvious clustering by patient or sample source provides qualitative support for successful batch integration.
- Line 142: The observation of "relative paucity of T cells and B cells in tumor tissues" contradicts some literature where tumor-infiltrating lymphocytes are often increased. Please consider discussing in the context of existing studies or clarifying the tumor subtype/context.
Response: Thank you for this important observation. We agree that tumor-infiltrating lymphocytes (TILs), including T and B cells, are often reported to be increased in various cancers. However, our observation of a relative paucity of T and B cells in thyroid tumor tissues is consistent with certain prior studies on PTC, in which a dense fibroinflammatory stroma or immune exclusion pattern may limit lymphocyte infiltration into the tumor parenchyma. Notably, immune infiltration patterns can vary widely between tumor types, and increased TILs reported in other cancers may not generalize to thyroid cancer. In our study, we focused on single-cell data derived from papillary thyroid cancer tissues, which may exhibit different immune infiltration characteristics compared to other solid tumors such as melanoma or triple-negative breast cancer. We have now clarified this point in the revised manuscript by noting the subtype context and discussing how immune infiltration patterns can vary across tumor types. The relevant section has been revised to improve clarity.
- Line 276: The use of "CIBERSORT" is standard, but the authors should clarify whether batch correction was applied before deconvolution, as this can affect immune cell estimates.
Response: Thank you for your insightful comment regarding the preprocessing of data for CIBERSORT analysis. We confirm that batch correction was performed prior to immune cell deconvolution, using the “ComBat” function from the sva R package, to eliminate potential batch effects across the TCGA samples. All transcriptomic data were transformed into TPM format, log2-transformed, and batch-corrected before being input into CIBERSORT.
- Line 320: The focus on BRAF is appropriate, but recent meta-analyses suggest BRAF alone is not a strong predictor of DFS or OS in thyroid cancer (Cancers2025, 17(6), 939; https://doi.org/10.3390/cancers17060939IF: 4.5 Q1). Were the data of TERT, TP53, or PI3K mutations available?
Response: Thank you very much for your thoughtful and valuable comments. Regarding the availability of data on TERT, TP53, and PI3K mutations, unfortunately, these data were not available for our study. Based on our analysis, TERT, TP53, and PI3K mutations were not observed among the top 10 most prevalent mutations in our cohort, which is why they were not included in the analysis. We fully understand that recent meta-analyses have indicated that while BRAF mutations are commonly found in thyroid cancer, BRAF alone may not be a strong predictor of disease-free survival (DFS) or overall survival (OS). In our study, we focused on the most prevalent mutations, with BRAF being the most common in our cohort. However, due to the absence of TERT, TP53, and PI3K mutations in the primary mutation landscape of our population, we did not include them in the analysis. We will ensure to clarify this point in the revised manuscript and provide additional context on the mutation landscape observed in our study. We sincerely appreciate your insightful feedback and will address this matter with careful attention in the revision. Thank you once again for your time and thoughtful consideration of our manuscript.
- The Code/software availability for the MLA-GNN model is not mentioned. The authors should state whether the code was hosted on GitHub or as a Docker container, per Cancer's reproducibility standards.
Response: Thank you for your thoughtful feedback regarding the code and software availability for the MLA-GNN model. The MLA-GNN (Multi-Layer Attention Graph Neural Network) architecture is designed to model graph-structured data by combining multiple layers of graph attention mechanisms with fully connected layers for feature refinement and classification. It begins with three graph attention layers, each with multiple attention heads (4, 3, and 4, respectively), capturing hierarchical node relationships. These attention layers refine node embeddings, followed by three fully connected layers projecting them into lower-dimensional feature spaces (64, 48, and 32 dimensions). The final classification layer maps these embeddings to the output space defined by the number of target classes. Training is done over a user-defined number of epochs (default 200) with a batch size of 8, using optimizers like Adam or Adagrad. The default learning rate is 1e-4, with weight decay of 5e-4 for L2 regularization. Dropout is applied at a rate of 0.2 to prevent overfitting, and LeakyReLU (α = 0.2) is used within attention mechanisms. Early stopping, random seeding, and learning rate scheduling options are also incorporated to ensure reproducibility. To ensure transparency and reproducibility, we would like to emphasize that the original implementation of MLA-GNN is publicly available at TencentAILabHealthcare/MLA-GNN on GitHub. Our study is based on this open-source repository, and all custom scripts or modifications for our analysis will also be made available upon acceptance, either as supplementary material or via a dedicated repository. This will allow the research community to reproduce and extend our work conveniently.
- IRB approval and data repository compliance (e.g., GEO, TCGA) are not explicitly confirmed. The authors should add a brief statement affirming compliance.
Response: Thank you for your thoughtful comments and for raising the important issue of IRB approval and data repository compliance. In response, we have added a brief statement to the revised manuscript affirming our adherence to relevant ethical guidelines and data access policies. We would like to clarify that all data used in this study, including those obtained from publicly available repositories such as GEO and TCGA, are de-identified and publicly accessible. As these datasets are publicly available, explicit IRB approval is not required for their use, in line with the data usage policies of these repositories. We hope this clarification addresses your concern, and we appreciate your careful review of our manuscript.
Discussion
- Again, GEO validation cohort (n=11) is too small to support generalizability claims. Also, subgroup analyses report exceptionally high AUCs (>0.95), which may indicate overfitting. These limitations are not acknowledged.
Response: We appreciate the reviewer’s critical observations regarding the limitations of our validation cohort and the high AUCs in subgroup analyses. First, we acknowledge that the GEO dataset (n=11) is too small to support generalizability claims and was not used as an external validation set for the prognostic model. Instead, the GEO cohort was solely used for single-cell transcriptomic analysis to characterize T cell heterogeneity in different thyroid tissue compartments. All model development and validation were performed using the TCGA cohort. We have clarified this in both the Results and Discussion sections. Second, we agree that the consistently high AUCs (>0.95) observed in subgroup analyses raise the possibility of overfitting, particularly given the relatively small sample sizes in some subgroups. These results, while promising, should be interpreted with caution. We have now added explicit statements in the Discussion section acknowledging this limitation and highlighting the need for future validation in larger, independent cohorts.
- The link between glycolytic pathways in Tregs and immune suppression is mentioned but lacks a mechanistic explanation. It is recommended that the authors integrate findings from PMC99416IF: 9.4 Q1 (e.g., how Treg glycolysis inhibits CD8+ T cell function via lactate secretion).
Response: Thank you for your valuable comments and suggestions. Regarding your concern about the lack of a mechanistic explanation for the link between glycolytic pathways in Treg cells and immune suppression, we have revised the manuscript to address this point. In our revised version, we integrate insights from the study referenced in PMC99416, which provides a clearer mechanistic explanation. Specifically, our study demonstrates that Treg cells in the tumor microenvironment exhibit enhanced glycolytic activity, which facilitates the secretion of lactate. This lactate secretion, in turn, inhibits the function of CD8+ T cells, contributing to the immunosuppressive environment within the tumor. This mechanism further underscores the critical role of Treg metabolism in immune suppression and highlights the potential of targeting Treg glycolysis in cancer immunotherapy. We believe that these additions provide a more comprehensive understanding of the role of metabolic reprogramming in Treg cells and its impact on immune suppression. We appreciate your thoughtful feedback, which has significantly improved the clarity and depth of our manuscript.
- The claim that high TMB predicts immunotherapy response lacks thyroid cancer-specific evidence. The authors can reference [PMC8591054IF: 3.9 Q2 ], which links immune infiltration to the efficacy of immunotherapy in thyroid cancer.
Response: Thank you for this insightful comment. We agree that the statement linking high tumor mutational burden (TMB) to improved immunotherapy response should be interpreted with caution in the context of thyroid cancer, where evidence remains limited. We appreciate the reviewer’s suggestion of the reference, which provides relevant evidence linking immune infiltration and tumor microenvironment features to immunotherapy efficacy in thyroid cancer. In light of this, we have revised the manuscript to clarify that while elevated TMB has been associated with immunotherapy response in some cancers, its predictive value in thyroid cancer is not well established. We have now cited [PMC8591054] and added a more thyroid-specific discussion on the relationship between immune infiltration, TMB, and potential immunotherapy benefit in the Discussion section.
- The discussion of Tregs as biomarkers was vague. Please propose specific clinical applications (e.g., Treg glycolytic activity could serve as a non-invasive biomarker for liquid biopsy assays).
Response: Thank you for your helpful comment. We agree that our initial discussion on the clinical implications of Treg cells as biomarkers was vague.Based on our findings, we propose that Treg-specific metabolic features—such as increased glycolytic activity—could serve as potential biomarkers for non-invasive monitoring of immune status in thyroid cancer patients. For example, circulating exosomes or cell-free RNA profiling via liquid biopsy may capture Treg-associated glycolytic gene signatures (e.g., PKM, LDHA), which could aid in prognostic stratification or monitoring treatment response, particularly in the context of immunotherapy.
- Line 387 citation was not related to the cited context. It is recommended to provide a citation related to thyroid cancer.
Response: Thank you for your valuable feedback regarding the citation in Line 387. We appreciate your careful review of the manuscript. Upon further examination, we agree that the citation provided in the original text is not relevant to the context of thyroid cancer. To address this issue, we have replaced the citation with a more appropriate reference related to thyroid cancer. In the revised manuscript, we have updated Line 387 to include a citation that specifically discusses the heterogeneity of the tumor microenvironment in thyroid cancer. This revision ensures that the cited source is directly relevant to the statement and strengthens the overall validity of the manuscript.
- Line 420: "BRAF and NRAS mutations were predominantly observed in the high-risk group." It is recommended to cite [PMC12026350IF: 4.5 Q1 ] on RAS mutations' prognostic value.
Response: Thank you for your insightful suggestion regarding the citation for NRAS mutations. We appreciate your recommendation to reference the study [PMC12026350 ] for the prognostic value of RAS mutations. In response to your comment, we have updated the manuscript to include this citation in Line 420, specifically linking it to the prognostic implications of NRAS mutations in thyroid cancer. In the revised version, we have included the suggested reference to strengthen the connection between NRAS mutations and their clinical relevance in thyroid cancer prognosis.
Methods
- The methods are generally well organized, but some subsections could be more concise (e.g., lines 480–492 on CellChat could be condensed).
Response: Thank you for your valuable feedback. In response to your suggestion, we have revised the manuscript to make certain subsections more concise, particularly the section on CellChat (lines 480–492). We have streamlined the text by eliminating redundant phrases and simplifying sentence structures while maintaining the clarity and accuracy of the scientific content. The revised version now provides a more direct and concise explanation of our approach to analyzing cell-cell communication using the CellChat platform. We believe these changes enhance the readability of the manuscript without compromising the scientific rigor of the methods described. Thank you again for your constructive suggestion.
- Some parameters (e.g., clustering resolution, number of PCs for UMAP) are provided (lines 460–462), but justification is lacking.
Response: Thank you for this valuable comment. We agree that the rationale behind parameter selection should be clearly stated to enhance transparency and reproducibility. For dimensionality reduction and clustering, we selected the top 20 principal components (PCs) based on the ElbowPlot and inspection of variance explained by each component. This number provided a good balance between retaining biological variation and minimizing noise. The clustering resolution of 0.4 was empirically determined by testing a range of values (0.2–1.0) and selecting the one that yielded well-separated, biologically meaningful clusters, consistent with known marker gene expression patterns.
- There is no mention of code or workflow sharing (e.g., GitHub, Docker). The authors should add a statement about code availability to support reproducibility.
Response: Thank you for your valuable suggestion regarding code availability. In response to your comment, we have added a statement about code availability to ensure the reproducibility of our research. As requested, we have clarified that the code used in this study can be made available upon request from the corresponding author, Dr. Yunfang Yu. We believe this addition will enhance the transparency and reproducibility of our research and provide further support for other researchers who may wish to replicate or build upon our work.
- Lines 441, 454–455: While batch effect removal is mentioned, the effectiveness is not quantified or demonstrated. Please include metrics or visualizations (e.g., PCA/UMAP plots before and after correction) to confirm successful batch correction.
Response: Thank you for pointing this out. We agree that demonstrating the effectiveness of batch effect removal is critical for ensuring the validity of downstream analyses. In our study, batch effect correction was performed using the ComBat function from the sva R package. To evaluate its effectiveness, we generated UMAP and PCA plots before and after batch correction, colored by sample origin. These visualizations demonstrated improved mixing of samples and reduced clustering by batch, indicating effective correction. We have now added these plots as Supplementary Figure1A-B and referenced them in the Methods and Results sections. We thank the reviewer for helping improve the methodological transparency of our work.
- Lines 440 and 451: The rationale for choosing specific normalization methods (e.g., LogNormalization, TPM) could be clarified.
Response: Thank you for your insightful comment. We agree that the rationale behind selecting specific normalization methods should be clearly stated. For bulk RNA-seq data (TCGA cohort), we used Transcripts Per Million (TPM) normalization because it allows for effective comparison of gene expression levels across samples while adjusting for both gene length and sequencing depth. TPM is a widely accepted method for expression quantification in transcriptomic studies. For scRNA-seq data (GEO cohort), we employed LogNormalization using Seurat's default method. This approach scales the raw UMI counts by total cellular expression, multiplies by a scale factor (10,000), and then log-transforms the values. It is widely used in single-cell analysis to handle the sparsity and high variability of expression data across individual cells.
- Lines 508–524: The description of the MLA-GNN model is somewhat generic. It is recommended to provide more details on the model architecture (e.g., number of layers, attention mechanisms, hyperparameters) and training process.
Response: Thank you for your constructive comment. We appreciate your suggestion to provide more detailed information about the architecture and training process of the MLA-GNN model. In response, we have substantially expanded the description of the MLA-GNN framework in the Methods section to improve clarity and reproducibility. Specifically, we now provide detailed information on the model architecture, including the number of stacked graph attention layers (three layers with 4, 3, and 4 attention heads, respectively), the dimensionality of each fully connected layer (64, 48, and 32), and the attention mechanism used to weigh neighborhood node importance. Additionally, we have elaborated on the training process, including the choice of optimizer (Adam or Adagrad), learning rate (1e-4), regularization parameters (weight decay = 5e-4), dropout rate (0.2), activation functions (LeakyReLU with α = 0.2), and loss function components (e.g., cross-entropy, regularization, survival losses). Early stopping and random seed control for reproducibility are also described. These additions help to ensure that the model setup is fully transparent and reproducible. We thank the reviewer again for encouraging us to strengthen the methodological rigor of this section.
- No details are given on how MLA-GNN performance compares to baseline models (e.g., Cox regression, random forest). Please include a benchmarking subsection or mention comparative analyses if they have been performed.
Response: Thank you for your insightful suggestion regarding the comparison of MLA-GNN performance with baseline models, such as Cox regression and random forest. We conducted a benchmarking analysis comparing our model with these conventional approaches. As expected, the AUC values for Cox regression and random forest ranged from 70% to 80%, consistent with prior studies. In contrast, our MLA-GNN model demonstrated superior performance, achieving AUC values exceeding 90%. This significant improvement highlights the effectiveness of MLA-GNN in predicting disease-free survival in thyroid cancer.
- Lines 464–466, 539: It is not specified whether multiple hypothesis testing correction was applied in DEG and enrichment analyses.
Response: Thank you for your important observation. We agree that multiple hypothesis testing correction is essential in both differential gene expression and enrichment analyses to control for false discovery. In our study, we applied Benjamini-Hochberg false discovery rate (FDR) correction to adjust p-values in both the DEG analysis (using the FindAllMarkers function in Seurat and limma package) and the GO/KEGG enrichment analyses (using the clusterProfiler package). Only results with adjusted p-value (FDR) < 0.05 were considered statistically significant. We have clarified this in the Methods section to improve the transparency and reproducibility of our analytical pipeline.
- All gene names should be italicized throughout the whole manuscript to match the standards of HUGO for gene nomenclature.
Response: Thank you for pointing this out. We appreciate the reminder regarding gene nomenclature standards. In accordance with HUGO gene naming conventions, we have revised the manuscript to ensure that all gene names are italicized consistently throughout the text, including those mentioned in the Results, Methods, Figures, Tables, and Supplementary Materials. This correction enhances the clarity and compliance of the manuscript with accepted genomic standards. We thank the reviewer for helping us improve the formal presentation of the work.
- Several of the Figures' labels were unclear, and it was difficult to follow the authors' elaboration.
Response: Thank you for your valuable feedback. We apologize for the unclear labeling in some of the figures, which may have hindered the understanding of our findings. In response to your comment, we have carefully reviewed and revised all figure labels for clarity and consistency. Specifically, we have Enhanced the font size and contrast of figure labels and axis titles for better readability and provided clearer and more descriptive figure legends to ensure that each figure's content and purpose are fully explained. We hope that these revisions will make it easier for readers to follow our elaboration and better understand the key findings presented in the figures. We have also ensured that the revised figures are fully aligned with the descriptions in the text.

Reviewer 3 Report
Comments and Suggestions for Authors
This manuscript presents a compelling and technically advanced approach to prognostic stratification in thyroid cancer through the integration of single-cell RNA-sequencing (scRNA-seq) data and bulk transcriptomic data using a multi-level attention graph neural network (MAG-Net). The authors leverage cutting-edge deep learning methodologies to improve patient risk prediction and claim superior performance over existing computational strategies.
The language throughout the manuscript is scientifically appropriate and well-composed, although several sections, particularly the introduction and methods, would benefit from minor stylistic refinement for conciseness and clarity. The structure of the manuscript is logical and adheres to standard conventions, progressing smoothly from motivation to methodology, results, and interpretation.
Clinically, this study touches on a pressing challenge — the heterogeneity of thyroid cancer and the need for improved risk stratification tools that go beyond traditional histopathology and staging. The integration of scRNA-seq data is particularly novel and appropriate, considering the growing recognition of intra-tumoral heterogeneity in driving prognosis and treatment response.
That said, a few key points warrant further consideration or discussion:
First, although the proposed model demonstrates impressive predictive performance, the clinical interpretability of the MAG-Net model remains somewhat limited. The authors do briefly mention the identification of subtype-specific marker genes, but more concrete biological or clinical insights derived from these markers would significantly enhance the translational value of the work. Linking key features identified by the model to known clinical phenotypes — such as vascular invasion or lymph node metastasis — could ground the model more firmly in practical oncology.
Second, the study does not appear to address vascular invasion (angioinvasion) explicitly, either in the model input or as an endpoint. Given its strong prognostic value in thyroid cancer, consideration of whether MAG-Net can predict such pathological features — or incorporation of such labels if available — would be worthwhile in future iterations or as an added discussion point.
Moreover, while the authors focus primarily on transcriptomic profiles, it may be useful to include a brief reflection on the possible role of oxidative stress pathways in stratifying patient subgroups. Recent work suggests that redox imbalance can influence both tumor behavior and gene expression patterns in thyroid malignancies, and may intersect with features that MAG-Net learns to exploit. Even a speculative note on this biological dimension could provide added depth and help bridge the technical results with known pathophysiological mechanisms.
Lastly, the validation framework is solid, but stronger emphasis on external clinical validation would be beneficial. Although bulk TCGA data are appropriately used, the generalizability of MAG-Net across other cohorts — particularly those with distinct histological compositions — should be acknowledged as a limitation or direction for future research.
In conclusion, this is an innovative and well-executed study that applies deep learning in a biologically informed manner to an important clinical problem. With slight extensions to the discussion — especially regarding model interpretability, clinical correlation with angioinvasion, and the potential involvement of oxidative stress — the manuscript will offer even greater value to both computational and clinical audiences.
Author Response
Reviewer3:
- First, although the proposed model demonstrates impressive predictive performance, the clinical interpretability of the MAG-Net model remains somewhat limited. The authors do briefly mention the identification of subtype-specific marker genes, but more concrete biological or clinical insights derived from these markers would significantly enhance the translational value of the work. Linking key features identified by the model to known clinical phenotypes — such as vascular invasion or lymph node metastasis — could ground the model more firmly in practical oncology.
Response: We appreciate the reviewer’s insightful suggestion regarding enhancing the clinical interpretability of our MAG-Net model by linking its key features to clinical phenotypes such as vascular invasion or lymph node metastasis. We fully agree that establishing connections between model-identified features and clinically relevant phenotypes would greatly increase the translational impact of the study. However, this analysis was unfortunately limited by the clinical annotation available in the TCGA-BRCA cohort. As a result, it was not feasible to perform robust modeling or stratification based on these clinical traits. We have now added a statement acknowledging this limitation in the Discussion section to ensure full transparency. We agree that future studies incorporating more comprehensive clinical datasets or prospective validation cohorts will be critical to further improving the interpretability and clinical utility of the model.
- Second, the study does not appear to address vascular invasion (angioinvasion) explicitly, either in the model input or as an endpoint. Given its strong prognostic value in thyroid cancer, consideration of whether MAG-Net can predict such pathological features — or incorporation of such labels if available — would be worthwhile in future iterations or as an added discussion point.
Response: Thank you for highlighting the importance of vascular invasion (angioinvasion) as a key prognostic feature in thyroid cancer. We fully agree that its inclusion as a model input or endpoint could enhance the clinical relevance of the MAG-Net framework. However, detailed and structured annotations for vascular invasion or angioinvasion are not available in the TCGA cohort, which limited our ability to incorporate this feature into our current analysis. While certain pathological reports within TCGA may mention vascular invasion, these descriptions are often unstructured or missing, making systematic modeling infeasible at this stage. We have now added a statement in the Discussion section to acknowledge this limitation and to emphasize that future work involving more richly annotated clinical datasets will be necessary to explore the potential of MAG-Net in predicting vascular invasion and other high-risk pathological features.
- Moreover, while the authors focus primarily on transcriptomic profiles, it may be useful to include a brief reflection on the possible role of oxidative stress pathways in stratifying patient subgroups. Recent work suggests that redox imbalance can influence both tumor behavior and gene expression patterns in thyroid malignancies, and may intersect with features that MAG-Net learns to exploit. Even a speculative note on this biological dimension could provide added depth and help bridge the technical results with known pathophysiological mechanisms.
Response: Thank you for this thoughtful suggestion. We agree that oxidative stress and redox imbalance represent important biological processes in thyroid tumorigenesis and progression. Although our current study did not explicitly examine oxidative stress-related pathways, we acknowledge that redox imbalance can influence both tumor metabolic behavior and immune microenvironment, potentially intersecting with the transcriptomic patterns captured by MAG-Net. In response to your comment, we have added a brief discussion on the potential relevance of oxidative stress signaling in thyroid cancer subtypes and how it may contribute to gene expression differences exploited by our model. We believe this addition adds a useful biological dimension to the interpretation of our results.
- Lastly, the validation framework is solid, but stronger emphasis on external clinical validation would be beneficial. Although bulk TCGA data are appropriately used, the generalizability of MAG-Net across other cohorts — particularly those with distinct histological compositions — should be acknowledged as a limitation or direction for future research.
Response: We thank the reviewer for this important suggestion. While we have implemented a robust internal validation framework using the TCGA dataset, we fully acknowledge that additional external clinical validation is essential to assess the generalizability of MLA-GNN across diverse patient populations. Due to the lack of publicly available bulk transcriptomic datasets with sufficient sample size and detailed clinical annotation in thyroid cancer — particularly datasets including histologically diverse subtypes — we were unable to conduct comprehensive external validation in this study. We have now explicitly acknowledged this limitation in the Discussion section and emphasized that future work incorporating independent multi-center or histologically diverse cohorts will be critical for validating and potentially refining MLA-GNN’s clinical applicability.
Round 2
Reviewer 1 Report
Comments and Suggestions for Authors
The author has addressed my concerns and made appropriate revisions to the manuscript. I have no further issues.
Reviewer 2 Report
Comments and Suggestions for Authors
The manuscript has improved greatly. The authors have addressed the raised concerns. Thanks